



# Organic nitrate chemistry and its implications for nitrogen budgets in an isoprene- and monoterpene-rich atmosphere: constraints from aircraft (SEAC[4]RS) and ground-based (SOAS) observations in the Southeast US

J. A. Fisher[1,2], D. J. Jacob[3,4], K. R. Travis[3], P. S. Kim[4], E. A. Marais[3], C. Chan Miller[4], K. Yu[3], L. Zhu[3], R. M. Yantosca[3], M. P. Sulprizio[3], J. Mao[5,6], P. O. Wennberg[7,8], J. D. Crounse[7], A. P. Teng[7], T. B. Nguyen[7,a], J. M. St. Clair[7,b], R. C. Cohen[9,10], P. Romer[9], B. A. Nault[10,c], P. J. Wooldridge[9], J. L. Jimenez[11,12], P. Campuzano-Jost[11,12], D. A. Day[11,12], P. B. Shepson[13,14], F. Xiong[13], D. R. Blake[15], A. H. Goldstein[16,17], P. K. Misztal[16], T. F. Hanisco[18], G. M. Wolfe[18,19], T. B. Ryerson[20], A. Wisthaler[21,22], and T. Mikoviny[21]

[1]Centre for Atmospheric Chemistry, School of Chemistry, University of Wollongong, Wollongong, NSW, Australia
[2]School of Earth and Environmental Sciences, University of Wollongong, Wollongong, NSW, Australia
[3]Harvard John A. Paulson School of Engineering and Applied Sciences, Harvard University, Cambridge, MA, USA
[4]Department of Earth and Planetary Sciences, Harvard University, Cambridge, MA, USA
[5]Program in Atmospheric and Oceanic Sciences, Princeton University, Princeton, NJ, USA
[6]Geophysical Fluid Dynamics Laboratory/National Oceanic and Atmospheric Administration, Princeton, NJ, USA
[7]Division of Geological and Planetary Sciences, California Institute of Technology, Pasadena, CA, USA
[8]Division of Engineering and Applied Science, California Institute of Technology, Pasadena, CA, USA
[9]Department of Chemistry, University of California at Berkeley, Berkeley, CA, USA
[10]Department of Earth and Planetary Science, University of California at Berkeley, Berkeley, CA, USA
[11]Department of Chemistry and Biochemistry, University of Colorado, Boulder, CO, USA
[12]Cooperative Institute for Research in Environmental Sciences, University of Colorado, Boulder, CO, USA
[13]Department of Chemistry, Purdue University, West Lafayette, IN, USA
[14]Department of Earth, Atmospheric and Planetary Sciences, Purdue University, West Lafayette, IN, USA
[15]Department of Chemistry, University of California Irvine, Irvine, CA, USA
[16]Department of Environmental Science, Policy, and Management, University of California at Berkeley, Berkeley, CA, USA
[17]Department of Civil and Environmental Engineering, University of California at Berkeley, Berkeley, CA, USA
[18]Atmospheric Chemistry and Dynamics Laboratory, NASA Goddard Space Flight Center, Greenbelt, MD, USA
[19]Joint Center for Earth Systems Technology, University of Maryland Baltimore County, Baltimore, MD, USA
[20]Chemical Sciences Division, Earth System Research Lab, National Oceanic and Atmospheric Administration, Boulder, CO, USA
[21]Department of Chemistry, University of Oslo, Oslo, Norway
[22]Institute for Ion Physics and Applied Physics, University of Innsbruck, Innsbruck, Austria
[a]Now at Department of Environmental Toxicology, University of California at Davis, Davis, CA, USA
[b]Now at Atmospheric Chemistry and Dynamics Laboratory, NASA Goddard Space Flight Center, Greenbelt, MD, USA and Joint Center for Earth Systems Technology, University of Maryland Baltimore County, Baltimore, MD, USA
[c]Now at Department of Chemistry and Biochemistry and Cooperative Institute for Research in Environmental Sciences, University of Colorado, Boulder, CO, USA

*Correspondence to:* J.A. Fisher (jennyf@uow.edu.au)





**Abstract.** Formation of organic nitrates ($RONO_2$) during oxidation of biogenic volatile organic compounds (BVOCs: isoprene, monoterpenes) is a significant loss pathway for atmospheric nitrogen oxide radicals ($NO_x$), but the chemistry of $RONO_2$ formation and degradation remains uncertain. Here we implement a new BVOC oxidation mechanism (including updated isoprene chemistry, new monoterpene chemistry, and particle uptake of $RONO_2$) in the GEOS-Chem global chemical transport model

with $\sim25\times25$ km$^2$ resolution over North America. We evaluate the model using aircraft (SEAC$^4$RS) and ground-based (SOAS) observations of $NO_x$, BVOCs, and $RONO_2$ from the Southeast US in summer 2013. The updated simulation successfully reproduces the concentrations of individual gas- and particle-phase $RONO_2$ species measured during the campaigns. Gas-phase isoprene nitrates account for 25-50% of observed $RONO_2$ in surface air, and we find that another 10% is contributed by gas-phase monoterpene nitrates. Observations in the free troposphere show an important contribution from long-lived nitrates

derived from anthropogenic VOCs. During both campaigns, at least 10% of observed boundary layer $RONO_2$ were in the particle phase. We find that aerosol uptake followed by hydrolysis to $HNO_3$ accounts for 60% of simulated gas-phase $RONO_2$ loss in the boundary layer. Other losses are 20% by photolysis to recycle $NO_x$ and 15% by dry deposition. $RONO_2$ production accounts for 20% of the net regional $NO_x$ sink in the Southeast US in summer, limited by the spatial segregation between BVOC and $NO_x$ emissions. This segregation implies that $RONO_2$ production will remain a minor sink for $NO_x$ in the Southeast US

in the future even as $NO_x$ emissions continue to decline.

## 1   Introduction

Nitrogen oxide radicals ($NO_x \equiv NO + NO_2$) are critical in controlling tropospheric ozone production (Monks et al., 2015, and references therein) and influencing aerosol formation (Rollins et al., 2012; Ayres et al., 2015; Xu et al., 2015), with indirect impacts on atmospheric oxidation capacity, air quality, climate forcing, and ecosystem health. The ability of $NO_x$ to influence

ozone and aerosol budgets is tied to its atmospheric fate. In continental regions, a significant loss pathway for $NO_x$ is reaction with peroxy radicals derived from biogenic volatile organic compounds (BVOCs) to form organic nitrates (Liang et al., 1998; Browne and Cohen, 2012). $NO_x$ loss to organic nitrate formation is predicted to become increasingly important as $NO_x$ abundance declines (Browne and Cohen, 2012), as has occurred in the US over the past two decades (Hidy et al., 2014; Simon et al., 2015). Despite this increasing influence on the $NO_x$ budget, the chemistry of organic nitrates remains the subject of

debate, with key uncertainties surrounding the organic nitrate yield from BVOC oxidation, the recycling of $NO_x$ from organic nitrate degradation, and the role of organic nitrates in secondary organic aerosol formation (Paulot et al., 2012; Perring et al., 2013). Two campaigns in the Southeast US in summer 2013 provided datasets of unprecedented chemical detail for addressing these uncertainties: the airborne NASA SEAC$^4$RS (Studies of Emissions and Atmospheric Composition, Clouds, and Climate Coupling by Regional Surveys;  Toon et al., 2016) and the ground-based SOAS (Southern Oxidants and Aerosols Study). Here

we use a $\sim25\times25$ km$^2$ resolution 3-D chemical transport model (GEOS-Chem) to interpret organic nitrate observations from both campaigns, with focus on their impacts on atmospheric nitrogen (N) budgets.

Nitrogen oxides are emitted from natural and anthropogenic sources primarily as NO, which rapidly achieves steady state with $NO_2$. Globally, the dominant loss pathway for $NO_x$ is reaction with the hydroxyl radical (OH) to form nitric acid ($HNO_3$).





In the presence of VOCs, $NO_x$ can also be lost by reaction with organic peroxy radicals ($RO_2$) to form peroxy nitrates ($RO_2NO_2$) and alkyl and multifunctional nitrates ($RONO_2$) (O'Brien et al., 1995). Their daytime formation temporarily sequesters $NO_x$, facilitating its export to more remote environments (Horowitz et al., 1998; Paulot et al., 2012; Mao et al., 2013). $RO_2NO_2$ species are thermally unstable at boundary layer temperatures and decompose back to $NO_x$ on a time scale

of minutes, except for the longer-lived peroxyacylnitrates (PANs) (Singh and Hanst, 1981). $RONO_2$ species can dominate $NO_x$ loss when BVOC emissions are high and $NO_x$ emissions are low (Browne and Cohen, 2012; Paulot et al., 2012; Browne et al., 2014) and may be more efficient for reactive N export than PANs (Mao et al., 2013). The amount of $NO_x$ sequestered by $RONO_2$ depends on the interplay between BVOC and $NO_x$ emissions, the $RONO_2$ yield from BVOC oxidation, and the eventual $RONO_2$ fate. In industrialized regions, BVOC and $NO_x$ sources tend to be spatially segregated, limiting the impact

of $RONO_2$ production on $NO_x$ loss (Yu et al., 2016). However, the BVOC emission flux can be much larger than the $NO_x$ emission flux, so even with small yields $RONO_2$ may significantly impact the $NO_x$ budget (Browne and Cohen, 2012).

$RONO_2$ chemistry and impacts are illustrated schematically in Fig. 1, starting from reaction of $NO_x$ with BVOCs (mainly isoprene and monoterpenes) to form $RONO_2$. The $RONO_2$ yield ($\alpha$) from isoprene oxidation by OH has been inferred from laboratory and field experiments to be 4-15% (Tuazon and Atkinson, 1990; Chen et al., 1998; Sprengnether et al., 2002; Patchen

et al., 2007; Perring et al., 2009a; Paulot et al., 2009; Nguyen et al., 2014; Xiong et al., 2015). Models have shown nearly this full range of yields to be compatible with $RONO_2$ observations, depending on the chemical mechanism assumed. For example, two models using different isoprene reaction schemes both successfully reproduced observations from a 2004 aircraft campaign (ICARTT) - one assuming a 4% molar yield (Horowitz et al., 2007) and the other assuming an 11.7% molar yield (Mao et al., 2013). The $RONO_2$ yield from monoterpene oxidation by OH is even more uncertain. Laboratory measurements exist only

for $\alpha$-pinene, and these show divergent results: 26% (Rindelaub et al., 2015), 18% (Nozière et al., 1999), and 1% (Aschmann et al., 2002, a lower limit due to significant wall losses). $RONO_2$ yields remain a significant uncertainty in BVOC oxidation schemes, with implications for their impacts on $NO_x$ sequestration.

The fate of $RONO_2$ is of central importance in determining whether sequestered $NO_x$ is returned to the atmosphere or removed irreversibly. Many first generation $RONO_2$ (i.e., those formed from NO reaction with BVOC-derived peroxy radicals)

have a short lifetime against further oxidation to form a suite of second generation $RONO_2$ (Beaver et al., 2012; Mao et al., 2013; Browne et al., 2014), especially if they are produced from di-olefins such as isoprene or limonene. Laboratory studies indicate little $NO_x$ release during this process (Lee et al., 2014); however, $NO_x$ can be recycled by subsequent oxidation and photolysis of second generation species (Müller et al., 2014). Estimates of the $NO_x$ recycling efficiency, defined as the mean molar percentage of $RONO_2$ loss that releases $NO_x$, range from <5% to >50% for isoprene nitrates (INs) (Horowitz et al.,

2007; Paulot et al., 2009), and best estimates depend on assumptions about the IN yield (Perring et al., 2009a). $NO_x$ recycling efficiencies from monoterpene nitrates (MTNs) have not been observed experimentally, but model sensitivity studies have shown a 14% difference in boundary layer $NO_x$ between scenarios assuming 0% versus 100% recycling (assuming an initial 18% MTN yield, Browne et al., 2014). Uncertainty in the $NO_x$ recycling efficiency has a bigger impact on simulation of $NO_x$ and ozone than uncertainty in the $RONO_2$ yield (Xie et al., 2013).





Organic nitrates are typically more functionalized and less volatile than their BVOC precursors and are therefore more likely to partition to the particle phase. In the Southeast US, Xu et al. (2015) recently showed that particulate $RONO_2$ ($pRONO_2$) make an important contribution to total organic aerosol (5-12%), consistent with in situ observations from other environments (Brown et al., 2009, 2013; Fry et al., 2013; Rollins et al., 2012, 2013). Chamber experiments have shown high mass yields of aerosol from $NO_3$-initiated oxidation of isoprene (15-25%; Ng et al., 2008; Rollins et al., 2009) and some monoterpenes (33-65%; Fry et al., 2014). There is evidence that $RONO_2$ from OH-initiated oxidation also form aerosol, although with lower yields, possibly via multi-functionalized oxidation products (Kim et al., 2012; Lin et al., 2012; Rollins et al., 2012; Lee et al., 2014). $pRONO_2$ are removed either by deposition or by hydrolysis to form $HNO_3$ (Jacobs et al., 2014; Rindelaub et al., 2015). Both losses augment N deposition to ecosystems (Lockwood et al., 2008). Aerosol partitioning competes with photochemistry as a loss for gas-phase $RONO_2$ with impacts for $NO_x$ recycling. Partitioning also competes with gas-phase deposition, and because lifetimes against deposition are much longer for organic aerosols than for gas-phase precursors (Wainwright et al., 2012; Knote et al., 2015), this process may shift the enhanced N deposition associated with $RONO_2$ (Zhang et al., 2012; Nguyen et al., 2015) to ecosystems further downwind of sources.

The 2013 SEAC$^4$RS and SOAS campaigns provide a unique resource for evaluating the impact of BVOC-derived organic nitrates on atmospheric $NO_x$. Both campaigns provided datasets of unprecedented chemical detail, including isoprene, monoterpenes, total and particle-phase $RONO_2$, and speciated INs; during SOAS these were further augmented by measurements of MTNs. Continuous measurements from the SOAS ground site provide high temporal resolution and constraints on diurnal variability (e.g., Nguyen et al., 2015; Xiong et al., 2015). These are complemented by extensive boundary layer profiling across a range of chemical environments from the SEAC$^4$RS airborne measurements (Toon et al., 2016). Combined, the campaigns covered the summer period when BVOC emissions in the Southeast US are at a maximum (Palmer et al., 2006). These data offer new constraints for testing models of organic nitrate chemistry, with implications for our understanding of $NO_x$, ozone, and aerosol budgets in BVOC-dominated environments worldwide.

We examine here the impact of BVOC oxidation on atmospheric $NO_x$, using the 2013 campaign data combined with the GEOS-Chem model. The version of GEOS-Chem used in this work represents a significant advance over previous studies, with higher spatial resolution ($\sim$25$\times$25 km$^2$) that better captures the spatial segregation of BVOC and $NO_x$ emissions (Yu et al., 2016); updated isoprene nitrate chemistry incorporating new experimental and theoretical findings (e.g., Lee et al., 2014; Müller et al., 2014; Peeters et al., 2014; Xiong et al., 2015); addition of monoterpene nitrate chemistry (Browne et al., 2014; Pye et al., 2015); and consideration of particle uptake of gas-phase isoprene and monoterpene nitrates. We first evaluate the updated GEOS-Chem simulation using SOAS and SEAC$^4$RS observations of BVOCs, organic nitrates, and related species. We then use GEOS-Chem to quantify the fates of BVOC-derived organic nitrates in the Southeast US. Finally, we investigate the impacts of organic nitrate formation on the $NO_x$ budget.





## 2 Updates to GEOS-Chem simulation of organic nitrates

We use a new high resolution version of the GEOS-Chem CTM (www.geos-chem.org) v9-02, driven by assimilated meteorology from the NASA Global Modeling and Assimilation Office (GMAO) Goddard Earth Observing System Forward Processing (GEOS-FP) product. The model is run in a nested configuration (Wang et al., 2004), with native GEOS-FP horizontal reso-
lution of $0.25°$ latitude by $0.3125°$ longitude over North America (130-60°W, 9.75-60°N). Boundary conditions are provided from a $4° \times 5°$ global simulation, also using GEOS-Chem. The native GEOS-FP product includes 72 vertical layers of which $\sim$38 are in the troposphere. Temporal resolution of GEOS-FP is hourly for surface variables and 3-hourly for all others. Our simulations use a time step of 5 minutes for transport and 10 minutes for emissions and chemistry.

GEOS-Chem has been applied previously to simulation of organic nitrates in the Southeast US (e.g., Fiore et al., 2005;
Zhang et al., 2011; Mao et al., 2013). Mao et al. (2013) recently updated the GEOS-Chem isoprene oxidation mechanism to include explicit production and loss of a suite of second generation isoprene nitrates and nighttime oxidation by nitrate radicals. While their updated simulation showed good agreement with aircraft observations from the 2004 ICARTT campaign over the eastern US, we find that the more detailed chemical payloads available during SOAS and SEAC$^4$RS highlight deficiencies in that mechanism, resulting in large model biases in $RONO_2$.

A major component of this work is modification of the organic nitrate simulation in GEOS-Chem. Our focus here is on the BVOC-derived nitrates for which field measurements are newly available. GEOS-Chem simulation of PANs was recently updated by Fischer et al. (2014) and is not discussed here. Our improvements to the $RONO_2$ simulation are detailed below and include updates to isoprene oxidation chemistry, addition of monoterpene oxidation chemistry, and inclusion of aerosol uptake of $RONO_2$ followed by particle-phase hydrolysis. Other updates from GEOS-Chem v9-02 and comparison to Southeast
US observations are presented in several companion papers. Kim et al. (2015) describe the aerosol simulation and Travis et al. (2016) the gas-phase oxidant chemistry. Constraints on isoprene emissions from satellite formaldehyde observations are described by Zhu et al. (in preparation). The low-$NO_x$ isoprene oxidation pathway and implications for organic aerosols are described by Marais et al. (2016). Finally, Yu et al. (2016) evaluate the impact of model resolution and spatial segregation of $NO_x$ and BVOC emissions on isoprene oxidation.

### 2.1 Isoprene oxidation chemical mechanism

The basic structure of the GEOS-Chem isoprene oxidation mechanism is described by Mao et al. (2013), with updates to low-$NO_x$ pathways described and validated by Travis et al. (2016). Figure 2 shows our updated implementation of OH-initiated isoprene oxidation in the presence of $NO_x$ leading to isoprene nitrate (IN) formation. Isoprene oxidation by OH produces isoprene peroxy radicals ($ISOPO_2$) in either $\beta$- or $\delta$-hydroxy peroxy configurations depending on the location of OH addition.
In the presence of $NO_x$, $ISOPO_2$ reacts with NO to either produce $NO_2$ (the dominant fate; Perring et al., 2013) or form INs, with the yield of INs ($\alpha$) defined as the branching ratio between these two channels. Early laboratory measurements of $\alpha$ suggested an IN yield between 4.4-12% (Tuazon and Atkinson, 1990; Chen et al., 1998; Sprengnether et al., 2002; Patchen et al., 2007; Paulot et al., 2009; Lockwood et al., 2010). More recent experiments indicate continuing uncertainty in $\alpha$, with





a measured yield of $\alpha = 9 \pm 4\%$ from the Purdue Chemical Ionization Mass Spectrometer (CIMS; Xiong et al., 2015) and $\alpha = 13 \pm 2\%$ from the Caltech $CF_3O^-$ Time-of-Flight CIMS (CIT-ToF-CIMS; Teng et al., in preparation), despite excellent agreement during calibrated intercomparison exercises using one isoprene nitrate isomer (4,3 ISOPN). The sensitivity of the CIT-ToF-CIMS is similar for all isomers of ISOPN (Lee et al., 2014), while the Purdue instrument is less sensitive to the

major isomer (1,2 ISOPN) (Xiong et al., 2015). Here, we use a first generation IN yield of $\alpha = 9\%$, which we find provides a reasonable simulation of the SOAS observations and is also consistent with the SOAS box model simulations of Xiong et al. (2015). We discuss the model sensitivity to the choice of $\alpha$ in Sect. 3.

For the oxidation of isoprene by OH, the mechanism described in Mao et al. (2013) assumed a first generation IN composition of 40% $\beta$-hydroxyl INs ($\beta$-ISOPN) and 60% $\delta$-hydroxyl INs ($\delta$-ISOPN). However, new theoretical constraints show that under

atmospheric conditions, $\delta$-channel peroxy radicals are only a small fraction of the total due to fast redissociation of peroxy radicals that fosters interconversion between isomers and tends towards an equilibrium population with more than 95% $\beta$-isomers (Peeters et al., 2014). Using a simplified box model based on the extended Leuven Isoprene Mechanism LIM1, we found $\delta$-isomers were 4-8% of the total peroxy pool in representative Southeast US boundary layer conditions (temperature $\sim$295-300 K, $ISOPO_2$ lifetime $\sim$20-60 seconds). In what follows, we use an IN distribution of 90% $\beta$-ISOPN and 10% $\delta$-

ISOPN. Our box modeling suggests 10% is an upper limit for the $\delta$-ISOPN pool; however, we maintain this value as we find it necessary for simulation of species with predominantly $\delta$-pathway origins, including glyoxal and the second generation INs propanone nitrate (PROPNN) and ethanal nitrate (ETHLN).

First generation ISOPN isomers formed via OH oxidation of isoprene have a short photochemical lifetime against atmospheric oxidation (Paulot et al., 2009; Lockwood et al., 2010; Lee et al., 2014). Here we use updated reaction rate constants

and products from Lee et al. (2014) that increase the $\beta$-ISOPN+OH reaction by roughly a factor of two and decrease ozonolysis by three orders of magnitude (relative to the previous mechanism based on Lockwood et al., 2010; Paulot et al., 2009). Changes in $\delta$-ISOPN reaction rate constants are more modest but in the same direction. For both isomers, reaction with OH forms a peroxy radical ($ISOPNO_2$) along with a small (10%) yield of isoprene epoxy diols (Jacobs et al., 2014). Rate constants and products of the subsequent oxidation of $ISOPNO_2$ to form a suite of second generation INs follow the Lee et al. (2014)

mechanism. We explicitly simulate methylvinylketone nitrate (MVKN) and methacrolein nitrate (MACRN), which are primarily from the $\beta$-pathway; PROPNN and ETHLN, which are primarily from the $\delta$-pathway (and $NO_3$-initiated oxidation); and $C_5$ dihydroxy dinitrate (DHDN), formed from both isomers (Lee et al., 2014).

Isoprene reaction with $NO_3$ is the dominant isoprene sink at night and can also be significant during the day (Ayres et al., 2015), producing INs with high yield (Perring et al., 2009b; Rollins et al., 2009). This reaction can account for more than

20% of isoprene loss in some environments (Brown et al., 2009) and may explain 40-50% of total $RONO_2$ in the Southeast (Mao et al., 2013; Xie et al., 2013). The mechanism used here is identical to that described by Mao et al. (2013). Reaction of isoprene with $NO_3$ forms a nitrooxy peroxy radical ($INO_2$). Subsequent reaction of $INO_2$ with NO, $NO_3$, itself, or other peroxy radicals forms a first generation $C_5$ carbonyl nitrate (ISN1) with 70% yield, while reaction with $HO_2$ forms a $C_5$ nitrooxy hydroperoxide (INPN) with 100% yield. In this simplified scheme, we do not distinguish between $\beta$- and $\delta$- isomers

for ISN1 and INPN, nor do we include the $C_5$ hydroxy nitrate species recently identified in chamber experiments (Schwantes





et al., 2015). Mao et al. (2013) lumped all second generation nitrates derived from ISN1 and INPN into a single species ($R_4N_2$), but here we assume that the lumped species is PROPNN on the basis of recent chamber experiments that show PROPNN to be a high-yield photooxidation product of INs from $NO_3$-initiated oxidation (Schwantes et al., 2015). This effectively assumes instantaneous conversion of INs to PROPNN, a simplification that results in a shift in the simulated diurnal cycle of PROPNN

(see Sect. 3). We do not include here the nitrooxy hydroxyepoxide product recently identified by Schwantes et al. (2015).

Possible fates for second generation INs include further oxidation, photolysis, uptake to the aerosol phase followed by hydrolysis (Sect. 2.3), and removal via wet and dry deposition. Müller et al. (2014) show that photolysis is likely significantly faster than reaction with OH for carbonyl nitrates (e.g., MVKN, MACRN, ETHLN, PROPNN) due to enhanced absorption cross sections and high quantum yields caused by the proximity of the carbonyl group (a strongly absorbing chromophore) to

the weakly-bound nitrate group. Here we followed the methodology of Müller et al. (2014) to increase the absorption cross sections of the carbonyl INs. The new cross sections are 5-15 times larger than in the original model, which used the IUPAC-recommended tert-butyl nitrate cross section for all carbonyl nitrates (Roberts and Fajer, 1989). Peak midday photolysis rates now range from $\sim 3 \times 10^{-5}$ $s^{-1}$ (PROPNN) to $\sim 3 \times 10^{-4}$ $s^{-1}$ (MACRN).

Removal by dry deposition has been updated based on new observations from the SOAS ground site. The dry deposition

calculation is now constrained to match observed deposition velocities for ISOPN, MVKN, MACRN, and PROPNN (Nguyen et al., 2015; Travis et al., 2016), with all other $RONO_2$ deposition velocities scaled to that of ISOPN. Wet scavenging of gases is described in Amos et al. (2012) and has been modified here to use the same Henry's Law coefficients as for dry deposition. Aerosol partitioning is described in Sect. 2.3 below.

## 2.2  Monoterpene oxidation chemical mechanism

Monoterpene chemistry is not included in the standard GEOS-Chem gas-phase chemical mechanism. Here we implement a monoterpene nitrate scheme developed by Browne et al. (2014) that was built on the RACM2 chemical mechanism (Goliff et al., 2013) and evaluated using aircraft observations over the Canadian boreal forest (Browne et al., 2014). Our implementation is summarized in Fig. 3 and described briefly below, with the full mechanism available at http://wiki.seas.harvard.edu/geos-chem/index.php/Monoterpene_nitrate_scheme. We include two lumped monoterpene tracers: API representing monoterpenes with

one double bond ($\alpha$-pinene, $\beta$-pinene, sabinene, and $\Delta$-3-carene) and LIM representing monoterpenes with two double bonds (limonene, myrcene, and ocimene). Combined, these species account for roughly 90% of all monoterpene emissions (Guenther et al., 2012), and we neglect other terpenes here. During the day, LIM and API are oxidized by OH to form peroxy radicals. Subsequent reaction with NO forms first generation monoterpene nitrates with a yield of 18% (Nozière et al., 1999). These can be either saturated (MONITS) or unsaturated (MONITU), with precursor-dependent partitioning as shown in Fig. 3. For

all subsequent discussion, we refer to their sum MONIT = MONITU + MONITS.

At night, both LIM and API react with $NO_3$ to form a nitrooxy peroxy radical that either decomposes to release $NO_2$ or retains the nitrate functionality to form MONIT. The branching ratio between these two fates is 10% nitrate-retaining for API + $NO_3$ (Browne et al., 2014) and 50% nitrate-retaining for LIM + $NO_3$ (Fry et al., 2014). In Browne et al. (2014), the API + $NO_3$ reaction used the $\alpha$-pinene + $NO_3$ rate constant from the Master Chemical Mechanims (MCMv3.2). We have updated this rate

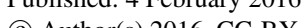
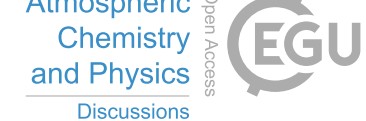

constant to $k_{API+NO_3} = 8.33 \times 10^{-13} e^{490/T}$, a rough average of the MCMv3.3 $\alpha$- and $\beta$-pinene values, as API comprises both $\alpha$- and $\beta$-pinenes (the dominant API components, present in roughly equal amounts during both SEAC[4]RS and SOAS). API and LIM also react with $O_3$, but this reaction does not lead to $RONO_2$ formation.

We do not distinguish between OH-derived and $NO_3$-derived MTN species. MONIT are subject to removal via wet and dry scavenging, aerosol uptake, photolysis, ozonolysis (MONITU only) and oxidation by OH. Here, we also add MONIT reaction with $NO_3$ with the same rate constant as used for nighttime isoprene nitrates. The products of MONIT oxidation are currently unknown; here we follow Browne et al. (2014) and assume oxidation produces a second generation monoterpene nitrate (HONIT) that undergoes dry deposition, photolysis, and oxidative loss. In our simulation, HONIT is also removed via aerosol uptake (Sect. 2.3).

## 2.3 Aerosol partitioning of $RONO_2$

Evidence from laboratory and field studies suggests aerosol uptake is a potentially significant loss pathway for gas-phase $RONO_2$ (e.g., Day et al., 2010; Rollins et al., 2010; Darer et al., 2011; Fry et al., 2013, 2014). In particular, BVOC oxidation by $NO_3$ radicals has been shown to result in high organic aerosol yields (Ng et al., 2008; Fry et al., 2009; Rollins et al., 2012). Recent work from SOAS highlighted the role of the monoterpenes + $NO_3$ reaction, with an estimated 23-44% yield of organic nitrate aerosol (Ayres et al., 2015) that can explain roughly half of nighttime secondary organic aerosol production (Xu et al., 2014). Isoprene + $NO_3$ results in smaller but still significant yields; Xu et al. (2014) estimate that isoprene was responsible for 20% of nighttime $NO_3$-derived organic aerosol observed during SOAS. Organic nitrate aerosol yields from daytime oxidation by OH are lower but non-negligible. At Bakersfield, for example, Rollins et al. (2013) found 21% of $RONO_2$ partitioned to the aerosol phase during the day, and that these could explain 5% of the total daytime organic aerosol mass.

Aerosol partitioning of $RONO_2$ has not previously been considered in GEOS-Chem. Here we add this process using a reactive uptake coefficient ($\gamma$) parameterization. Our parameterization was designed to provide a necessary sink for gas-phase $RONO_2$ species (overestimated in earlier iterations of our model), and therefore makes a number of simplifying assumptions. In particular, we do not allow $pRONO_2$ to re-partition to the gas phase (likely to impact the more volatile isoprene-derived nitrates), and uptake coefficients are defined to fit the measurements of gas-phase species. More accurate simulation of organic nitrate aerosols would require additional updates that take into account vapor pressure differences between species (as done recently by Pye et al., 2015) and incorporate new findings from SOAS (Ayres et al., 2015; Lee et al., 2015). For our simulation, we apply reactive uptake to all BVOC-derived $RONO_2$ except PROPNN and ETHLN, which lack hydroxyl groups and are therefore expected to be significantly less soluble. We assume an uptake coefficient of $\gamma$=0.005 for isoprene nitrates (from both daytime and nighttime chemistry) and $\gamma$=0.01 for all monoterpene nitrates. Our isoprene nitrate uptake coefficient is in the middle of the range predicted by Marais et al. (2016) using a mechanistic formulation, and is a factor of 4 lower than the upper limit for ISOPN inferred by Wolfe et al. (2015) using SEAC[4]RS flux measurements. Although simplified, we find this parameterization improves the model fit relative to the SEAC[4]RS and SOAS observations of individual gas-phase $RONO_2$ species measured by the CIT-ToF-CIMS and total $pRONO_2$ measured by an Aerosol Mass Spectrometer (AMS).





After partitioning to the aerosol, laboratory experiments have shown that $pRONO_2$ can hydrolyze to form alcohols and nitric acid via $pRONO_2 + H_2O \rightarrow ROH + HNO_3$. Some $pRONO_2$ species hydrolyze rapidly under atmospherically-relevant conditions, while others are stable against hydrolysis over timescales significantly longer than the organic aerosol lifetime against deposition (Darer et al., 2011; Hu et al., 2011; Liu et al., 2012; Jacobs et al., 2014; Rindelaub et al., 2015). Lifetimes against hydrolysis inferred from bulk aqueous and reaction chamber studies range widely from minutes (Darer et al., 2011; Rindelaub et al., 2015) to a few hours (Liu et al., 2012; Lee et al., 2015) to nearly a day (Jacobs et al., 2014). Here we apply a bulk lifetime of 1 hr against hydrolysis for the entire population of $pRONO_2$ (similar to Pye et al., 2015, who used a 3 hr bulk lifetime). In other words, our implementation of aerosol partitioning involves a two-step process of (1) uptake of gas-phase $RONO_2$ to form a simplified non-volatile $pRONO_2$ species, with rate determined by $\gamma$, followed by (2) hydrolysis of the simplified $pRONO_2$ species to form $HNO_3$, with rate determined by the lifetime against hydrolysis. In subsequent sections, we compare the simplified $pRONO_2$ formed as an intermediate during this process to total $pRONO_2$ derived from observations. The assumption of a single hydrolysis lifetime overestimates the loss of non-tertiary nitrates (Darer et al., 2011; Hu et al., 2011) and may lead to model bias in total $pRONO_2$, particularly in the free troposphere where the longer-lived species would be more prevalent (see Sect. 4). Our 1 hr bulk hydrolysis lifetime is shorter than the 2-4 hr lifetime found in recent analysis of SOAS data (Lee et al., 2015; Pye et al., 2015) - likely reflecting the simplifying assumptions of our uptake parameterization. In any case, the choice of hydrolysis lifetime does not affect the concentration of gas-phase $RONO_2$ species (because $pRONO_2$ cannot re-partition to the gas phase in the model), and we find this value provides a reasonable match to AMS measurements of total $pRONO_2$ at the surface during SOAS and SEAC[4]RS.

## 3   BVOCs and organic nitrates in the Southeast US

We evaluate the updated GEOS-Chem simulation using Southeast US measurements of isoprene, monoterpenes, and a suite of oxidation products from two field campaigns in summer 2013. SEAC[4]RS was a NASA aircraft campaign that took place in August-September 2013 (Toon et al., 2016). All observations discussed in this work were taken onboard the NASA DC-8 (data doi: 10.5067/Aircraft/SEAC[4]RS/Aerosol-TraceGas-Cloud), which was based in Houston, Texas with an ∼8-hour flight range. Understanding BVOC sources and chemistry was a primary goal of SEAC[4]RS, resulting in a large number of boundary layer flights over regions of enhanced biogenic emissions (Kim et al., 2015). Isoprene and monoterpene distributions in Southeast US surface air (80-94.5°W, 29.5-40°N, and below 1 km) measured by PTR-MS are shown in Fig. 4, and their campaign-median vertical profiles are shown in Fig. 5(b,c). Whole Air Sampler (WAS) measurements of isoprene and $\alpha$-pinene + $\beta$-pinene (Fig. S1) are similar, but with more limited sampling than the PTR-MS. All observations have been averaged to the spatial and temporal resolution of the model.

SOAS was a ground-based campaign that took place in June-July 2013 at the Centreville monitoring site near Brent, Alabama (32.903°N, 87.250°W). The site is located at the edge of a mixed coniferous and deciduous forest (Nguyen et al., 2015). Observations of isoprene and monoterpenes, measured by PTR-ToF-MS and averaged to hourly mean values, are shown in Fig. 6. Both species display a clear diurnal cycle with peak isoprene during day, reflecting the light- and temperature-dependent





source, and peak monoterpenes at night. For monoterpenes, the figure also shows the sum of $\alpha$-pinene + $\beta$-pinene as measured by 2D-GC-FID, which indicates that these are the dominant monoterpenes.

Figures 4, 5, and 6 compare observed BVOCs from both campaigns to the GEOS-Chem simulation, sampled to match the observations. Similar figures for $NO_x$ can be found in Travis et al. (2016) and in Fig. S2. Model bias relative to observations

is quantified using the normalized mean bias NMB$= 100\% \times [\sum_i (M_i - O_i) / \sum_i (O_i)]$, where $O_i$ and $M_i$ are the observed and modeled values and the summation is over all hours (SOAS) or unique gridbox-timestep combinations along the flight tracks (SEAC[4]RS). BVOC emissions are from MEGANv2.1 (Guenther et al., 2012) and have been decreased by 15% for isoprene and doubled for monoterpenes to better match aircraft (isoprene, monoterpene) and satellite (formaldehyde) observations (Kim et al., 2015; Zhu et al., in preparation). With these scalings applied, simulated surface isoprene and monoterpenes overestimate

somewhat the SEAC[4]RS data (Fig. 4, mainly due to a few simulated high-BVOC events), but the medians are well within the observed variability (Fig. 5). Model high bias above 500 m is likely caused by excessive vertical mixing through the simulated boundary layer (Travis et al., 2016). Relative to the SOAS data, simulated monoterpenes are biased low by a factor of two, while isoprene falls within the interquartile range of the measurements. The opposite sign of the SOAS monoterpene bias relative to the more spatially representative SEAC[4]RS data suggests a low bias in MEGANv2.1 monoterpene emissions that is

unique to the Centreville gridbox; errors in vertical mixing may also contribute. For isoprene, the model reproduces both the observed nighttime decline and the subsequent morning growth with a small delay ($\sim$1 hour).

The observed declines in isoprene at night (Fig. 6) and above the boundary layer (Fig. 5) reflect its short lifetime against oxidation. We find in the model that OH oxidation accounts for 90% of isoprene loss (Marais et al., 2016), but only 65% of monoterpenes loss (with $NO_3$ responsible for most of the rest). For isoprene, the subsequent fate of the peroxy radicals

(ISOPO$_2$) has been evaluated in detail by Travis et al. (2016), who also present an in-depth analysis of the $NO_x$ budget and impacts on ozone. They show that on average 56% of ISOPO$_2$ reaction during SEAC[4]RS is with NO, and that there is large spatial variability in this term that is accurately reproduced by the high-resolution GEOS-Chem simulation. Here we focus exclusively on this pathway and the resultant formation of $RONO_2$ from both isoprene and monoterpenes.

Observed near-surface mixing ratios of first generation isoprene nitrates (ISOPN) during SEAC[4]RS are shown in Fig. 7 and

are generally well represented by GEOS-Chem ($r = 0.61$; NMB = -1.2%). ISOPN vertical profiles in Fig. 5e indicate a rapid decline from the boundary layer to the free troposphere, reflecting the short atmospheric lifetime (2-4 hrs in our simulation; Table 1). Comparing the lowest altitude SEAC[4]RS observations to the SOAS median from the CIT-ToF-CIMS (black triangle) indicates an apparent vertical gradient from the surface to $\sim$500 m. This could be caused by spatial variability between the campaigns, or could reflect rapid dry deposition of ISOPN with limited vertical mixing. GEOS-Chem does not simulate this

SOAS-SEAC[4]RS difference, possibly due to overly strong vertical mixing through the modeled boundary layer as identified by Travis et al. (2016) from model comparison to SEACIONS ozonesonde observations.

During SOAS, ISOPN was measured simultaneously by the CIT-ToF-CIMS (Crounse et al., 2006; Nguyen et al., 2015) and the Purdue CIMS (Xiong et al., 2015), and Fig. 6 shows the diurnal cycles from both. Median ISOPN from the Purdue CIMS is a factor of two higher than that from the CIT-ToF-CIMS during daylight hours, with the most significant differences

in mid-late morning. In both datasets, ISOPN peaks around 10:00 am local time, is elevated until early evening, and declines





to a pre-dawn minimum. Simulated ISOPN from GEOS-Chem is in good agreement with the Purdue CIMS measurements except in the afternoon when modeled ISOPN shows a broad peak (rather than the observed decline) coincident with simulated peak isoprene (Fig. 6). After ~7:00 pm, the model captures the observed timing of the nighttime ISOPN decline seen in both datasets, as well as the rapid morning growth seen in the Purdue CIMS measurements.

As described in Sect. 2.1, there is considerable uncertainty in the ISOPN yield. We find here that a 9% yield provides the best simulation of the ensemble of SEAC$^4$RS and SOAS observations, given experimental constraints on oxidative loss rates (Lee et al., 2014) and dry deposition fluxes (Nguyen et al., 2015). Using model sensitivity studies, we found that applying a lower yield of 7% improved the agreement with the CIT-ToF-CIMS during SOAS, but worsened agreement with the other datasets and is inconsistent with the yields from laboratory experiments (Teng et al., in preparation). We also tested a higher

yield of 12%, and found the model overestimated observed SEAC$^4$RS and SOAS ISOPN (from both instruments) unless we invoked much larger aerosol uptake and/or added another ISOPN sink. ISOPN sinks (especially aerosol uptake) remain poorly constrained, and the uncertain parameter space describing these processes likely contains multiple solutions that fit the observations equally well (i.e., a higher yield could be accommodated by faster ISOPN loss to aerosol).

    Our finding that GEOS-Chem can reproduce the Purdue CIMS ISOPN observations using a 9% ISOPN yield is consistent

with the box model of Xiong et al. (2015). The chemical mechanisms used in both studies are similar. In both simulations, modeled ISOPN was overestimated unless an extra sink was included (also consistent with Wolfe et al., 2015, who inferred a missing sink based on SEAC$^4$RS flux measurements). While we assumed this sink was due to aerosol uptake, Xiong et al. (2015) invoked enhanced ISOPN photolysis. They argued that models typically underestimate the ISOPN absorption cross section by not taking into account the combined influence of the double bond and hydroxyl group in the ISOPN structure

(Fig. 2). Xiong et al. (2015) were better able to reproduce the observed ISOPN morning peak and afternoon decline when they increased the MCMv3.2 photolysis rate constant by a factor of 5. Including both faster ISOPN photolysis and uptake to the aerosol phase could be a means to accommodate a higher initial ISOPN yield, such as the 12-14% yield inferred from laboratory experiments with the CIT-ToF-CIMS (Teng et al., in preparation), although both sinks remain unverified. The nature of the sink has implications for $NO_x$ recycling from isoprene nitrates (photolysis recycles $NO_x$ while uptake removes it), and

this remains a source of uncertainty in our estimates of the impacts of $RONO_2$ on the $NO_x$ budget.

    Even more uncertain than ISOPN are the first generation monoterpene nitrates (MONIT). MONIT in GEOS-Chem is a lumped species that represents the sum of monoterpene nitrates from both daytime OH-initiated and nighttime $NO_3$-initiated oxidation (Sect. 2.2). The nighttime oxidation cascade involves a diversity of reactants (including NO, $HO_2$, $NO_3$, and other peroxy radicals) and produces a diversity of monoterpene nitrate species (Lee et al., 2015) that we do not distinguish here. In

the model, most MONIT is produced from the $NO_3$-initiated chemistry, resulting in mean MONIT concentrations of 30-60 ppt at night and ~10-20 ppt during the day.

    During SOAS, two monoterpene nitrates were measured by the CIT-ToF-CIMS: $C_{10}H_{17}NO_4$ and $C_{10}H_{17}NO_5$. We find that simulated MONIT shows the same diurnal pattern as the sum of the two measured species (with peak concentrations at night) but is a factor of 2-3 higher (Fig. S3). Pye et al. (2015) similarly found simulated MONIT was a factor of 7 higher

than observations using a version of the CMAQ model with explicit MONIT chemistry. The higher modeled values in both





studies presumably reflect inclusion in modeled MONIT of many species that were not measured by CIT-ToF-CIMS (including several identified during SOAS by Lee et al., 2015), as well as biases in the model mechanisms (most of the rate constants and products have not been measured). $NO_3$-initiated monoterpene oxidation is particularly uncertain and is likely too strong in GEOS-Chem, as indicated by large nighttime MONIT overestimates (Fig. S3) combined with monoterpene underestimates

(Fig. 6). Simulated nighttime peak values of $NO_3$-derived isoprene nitrates (ISN1) during SOAS are also up to a factor of 2 higher than the observations reported by Schwantes et al. (2015). This suggests that model biases in nighttime PBL heights and associated vertical mixing may also contribute to simulated nighttime overestimates for some $RONO_2$ species.

First generation ISOPN and MONIT undergo further oxidation to form a suite of second generation $RONO_2$ species that retain the nitrate functionality (Figs. 2, 3). Four of these species (MVKN, MACRN, PROPNN, and ETHLN) were measured

by the CIT-ToF-CIMS, with vertical profiles shown in Fig. 5 (f-h) and spatial distribution shown in Fig. 7. The model provides a good simulation of SEAC[4]RS MVKN+MACRN but underestimates the variability of PROPNN and ETHLN. In contrast, all three species show positive mean model biases relative to the SOAS surface observations. The model tends to overestimate PROPNN and ETHLN at night but underestimate them during the day (Fig. S3), reflecting the assumption in our mechanism that PROPNN is produced at night during $NO_3$-initiated isoprene oxidation. In reality, the nighttime chemistry produces

INs that only photo-oxidize to PROPNN after sunrise (Schwantes et al., 2015). This missing delay between nighttime $NO_3$ addition and subsequent daytime photo-oxidation likely also explains the model bias relative to the SEAC[4]RS observations, which mostly took place during daytime. Additional simplifications in the $NO_3$-initiated chemistry could also contribute to the biases, and preliminary simulations conducted with the AM3 model show that including more details of this chemistry improves model ability to match observed PROPNN (Li et al., in preparation). Some of the bias may also be due to error in

the assumed distribution between $\beta$- and $\delta$-channel OH-initiated oxidation, as both PROPNN and ETHLN are produced by the latter channel only.

The full time series of first and second generation INs measured at Centreville during SOAS are shown in Fig. 8. We also include the time series of observed particulate $RONO_2$ ($pRONO_2$) estimated from AMS measurements (Fry et al., 2013; Ayres et al., 2015; Lee et al., 2015; Day et al., in preparation) and of $\Sigma$ANs, the sum of all $RONO_2$ species (including

$pRONO_2$) as measured by thermal dissociation laser-induced fluorescence (TD-LIF; Day et al., 2002). Despite the biases identified above, the simulation captures the temporal variability in gas-phase, particulate, and total $RONO_2$ observed over the 6-week campaign, with correlation coefficients of $r \sim 0.6\text{-}0.7$. Low observed and modeled values for all species in early July (days 185-189) indicate suppressed BVOC emissions caused by low temperatures (Marais et al., 2016). The model underestimates both $pRONO_2$ and $\Sigma$ANs at night (Fig. S3), suggesting that hydrolysis of particulate monoterpene nitrates should be

slower than assumed here (Sect. 2.3).

The relationship between organic nitrates and formaldehyde (HCHO), a high-yield product of the $ISOPO_2$ + NO reaction, provides an additional test of the model chemistry and in particular the IN yield. Daytime isoprene oxidation in the presence of $NO_x$ co-produces HCHO and INs, resulting in an expected strong correlation between these species (Perring et al., 2009a). When INs dominate total $RONO_2$, the correlation should also be strong between HCHO and $\Sigma$ANs, and this relationship has

previously been used to constrain the IN yield when IN measurements were not available. For example, HCHO and $\Sigma$ANs





measurements from the 2004 ICARTT aircraft campaign showed moderate correlation with $r \sim 0.4\text{-}0.6$ (Perring et al., 2009a; Mao et al., 2013). However, linking the HCHO-$\Sigma$ANs correlation to the IN yield is complicated by the contribution to $\Sigma$ANs from other $RONO_2$ sources (e.g., monoterpene nitrates, anthropogenic nitrates, etc.). During SEAC$^4$RS, a better constraint can be obtained directly from the HCHO-IN relationship. Figure 9 shows the correlation between HCHO and the sum of ISOPN,

MVKN, and MACRN (we exclude PROPNN and ETHLN to avoid the biases identified previously). The figure shows the observed slope of 0.027 (ppt IN) (ppt HCHO)$^{-1}$ is reproduced by the model but with more scatter in the simulation ($r \sim 0.5$) than in the observations ($r \sim 0.7$). The similarity of the observed and simulated relationships in Fig. 9 lends confidence to the IN mechanism used here, at least for the $\beta$-peroxy channel.

## 4 Total alkyl and multifunctional nitrates ($\Sigma$ANs) and speciation

SEAC$^4$RS represents one of the first airborne campaigns to make measurements of individual BVOC-derived $RONO_2$ species. Without these speciated measurements, previous model evaluations of isoprene nitrate chemistry have relied on TD-LIF observations of $\Sigma$ANs (total $RONO_2$), with the assumption that gas-phase INs account for the majority of $\Sigma$ANs (Horowitz et al., 2007; Perring et al., 2009a; Mao et al., 2013; Xie et al., 2013). Figure 10a compares the TD-LIF $\Sigma$ANs measurement (solid line) to the sum of explicitly measured gas-phase $RONO_2$ species and total $pRONO_2$ (dashed line, combined CIT-ToF-CIMS,

WAS, and AMS measurements) during SEAC$^4$RS. The figure shows a large gap between measured $\Sigma$ANs and the total of speciated $RONO_2$ (including both gas-phase and aerosol contributions), especially near the surface ($\Sigma$ANs = 409 ppt, total speciated $RONO_2$ = 198 ppt). Figure 10a also shows the median surface $\Sigma$ANs measured during SOAS (198 ppt; black triangle). As for SEAC$^4$RS, SOAS total speciated $RONO_2$ is much lower (82 ppt) when calculated from the CIT-ToF-CIMS and AMS measurements. The gap is smaller, but still exists, when calculated using ISOPN from the Purdue CIMS (total $RONO_2$

= 102 ppt) or $pRONO_2$ from the TD-LIF (total $RONO_2$ = 139 ppt).

Some of the difference between the total speciated $RONO_2$ and $\Sigma$ANs measurements can be attributed to gas-phase nitrates not measured by CIT-ToF-CIMS or WAS. A number of these were identified during SOAS using a second ToF-CIMS operated by the University of Washington (Lee et al., 2015). In addition, SEAC$^4$RS observations of total $NO_y$ ($\equiv NO_x + HNO_3 + PAN + RONO_2$, including $pRONO_2$) are better balanced by including the $\Sigma$ANs than the speciated $RONO_2$ components ($\approx 81\%$ vs. 70% of

25 surface $NO_y$, compared to 56% with no $RONO_2$ contribution). Also contributing to the discrepancy are the large uncertainties still associated with $RONO_2$ measurement techniques. Lee et al. (2015) found that SOAS measurements of $pRONO_2$ differ by factors of 2-4, as also shown in Fig. S3, with the AMS lower than TD-LIF. Similarly, we showed in Sect. 3 that the two SOAS measurements of ISOPN differ by up to a factor of 2 (CIT-ToF-CIMS lower than Purdue CIMS, for reasons that remain unclear). Assuming similar uncertainties characterize the SEAC$^4$RS $RONO_2$ measurements, these could readily explain some

30 of the inability of the speciated measurements to close the $\Sigma$ANs budget in Fig. 10a.

Comparison of GEOS-Chem to the two total $RONO_2$ estimates in Fig. 10a shows that the model greatly underestimates SEAC$^4$RS $\Sigma$ANs relative to the TD-LIF measurement, with a much smaller underestimate relative to the speciated sum. The better fit to the speciated measurements than to the $\Sigma$ANs is consistent with the model's ability to match both individual gas-





phase $RONO_2$ species measured by the CIT-ToF-CIMS and total $pRONO_2$ measured by the AMS (Sect. 3). During SOAS, Fig. 8 shows that GEOS-Chem can reproduce much of the temporal variability in the $\Sigma$ANs ($r = 0.57$) with little bias.

Figure 10b compares the observed and simulated $RONO_2$ composition in the Southeast US during $SEAC^4RS$. For clarity, only the speciated measurements are shown in the figure. The observations show a constant 20-30 ppt background at all altitudes from small ($C_1$-$C_3$) $RONO_2$ produced from anthropogenic VOCs. The contributions of these small nitrates are consistent with the observed concentrations of their parent VOCs and with known reaction rate constants (Atkinson and Arey, 2003), $RONO_2$ yields (Perring et al., 2013), and $RONO_2$ lifetimes (Talukdar et al., 1997; Dahl et al., 2005; Worton et al., 2010) assuming steady state. GEOS-Chem does not simulate these nitrates under the assumption that their contributions to total $NO_y$ are insignificant. The $SEAC^4RS$ data clearly show that this assumption is not valid, at least for the US where natural gas production is a large alkane source, and is contributing to model bias in both $RONO_2$ and $NO_y$. Given the long lifetimes (weeks-months) of the small nitrates, the bias is particularly acute in the free troposphere and has implications for global N export.

In both observations and model, gas-phase INs (orange) account for half of speciated $RONO_2$ (25% of $\Sigma$ANs), split roughly equally between 1st and 2nd generation species. The model underestimates somewhat the 2nd generation INs, as seen previously in Figs. 5 and 7. In the model, gas-phase MTNs from monoterpenes (blue; not measured during $SEAC^4RS$) account for an additional 10% of simulated $RONO_2$ ($\sim$5% relative to $\Sigma$ANs). Previous studies during ICARTT also found a 10% MTN contribution to $RONO_2$ (Horowitz et al., 2007; Perring et al., 2009a), although MTNs have been neglected in more recent simulations (e.g., Mao et al., 2013; Xie et al., 2013). Other $C_4$-$C_5$ nitrates (yellow, including alkyl nitrates from WAS and alkene hydroxynitrates from CIT-ToF-CIMS) similarly contribute 5-10% of observed $RONO_2$; these are underestimated by 50% in GEOS-Chem because the model does not include nitrate formation from anthropogenic alkenes.

A significant fraction (10-20%) of $RONO_2$ was in the aerosol phase during $SEAC^4RS$. The model underestimates the observed $pRONO_2$ contribution in the free troposphere; however, some caution should be used when interpreting these data. Observed $pRONO_2$ is the product of measured total aerosol nitrate and the measured organic fraction of the nitrate aerosol, but during $SEAC^4RS$ the organic fraction was often not reported in the free troposphere due to interference from dust layers and instrumental issues. In these instances, the organic fraction of measured nitrate is assumed to be 0.8, based largely on surface measurements from multiple campaigns (Day et al., in preparation). In the free troposphere (>1.5 km), this assumption is applied to 85% of the $SEAC^4RS$ 1-minute data, and could lead to a high bias in the $pRONO_2$ observations. Nonetheless, it is also likely that simulated $pRONO_2$ is underestimated because of our assumption that all $pRONO_2$ species undergo rapid hydrolysis. In fact, many of the nitrates produced from BVOC oxidation are not expected to hydrolyze at all (Boyd et al., 2015; Pye et al., 2015) and so would have lifetimes sufficiently long for export out of the boundary layer.

Our simulated $RONO_2$ composition in Fig. 10b suggests a less important role for INs than identified from recent simulations of the ICARTT data. In an earlier version of GEOS-Chem, INs alone could explain all measured $\Sigma$ANs during ICARTT ($\sim$200 ppt at the surface; Mao et al., 2013), and both that model and a CMAQ simulation (Xie et al., 2013) suggested INs were dominated by 2nd generation species (70-90% of total INs). These earlier simulations did not account for either aerosol uptake and possible hydrolysis (Darer et al., 2011; Jacobs et al., 2014) or fast photolysis (Müller et al., 2014) of 2nd generation





INs, and so lifetimes were significantly longer than in our simulation. We performed sensitivity simulations without these additional IN sinks and found that the model overestimated observed 2nd generation INs by a factor of 3-5 during SEAC$^4$RS. It seems likely that 2nd generation IN overestimates in previous work were compensated for by omitting the contributions from MTNs and pRONO$_2$. Here, we find MTN and pRONO$_2$ combined contribute as much to total RONO$_2$ as either 1st or

2nd generation INs alone, and that excluding them would lead to major model shortcomings. The pRONO$_2$ contribution is especially important as different removal processes for gas-phase versus particulate species would have different implications for NO$_x$ budgets and N deposition.

## 5   Fate of organic nitrates and implications for nitrogen budgets

Table 1 summarizes the dominant fates and lifetimes of individual gas-phase RONO$_2$ in the Southeast US boundary layer

during the SEAC$^4$RS campaign (12 Aug - 23 Sep) as calculated from GEOS-Chem. The contribution of different fates to total gas-phase RONO$_2$ loss is illustrated in Fig. 11a. Loss processes that recycle RONO$_2$ by converting between RONO$_2$ species (e.g., from first to second generation) are not included. Total simulated RONO$_2$ loss is dominated by aerosol hydrolysis, with an additional large loss to deposition that is consistent with the rapid deposition fluxes of both INs and MTNs observed during SOAS (Nguyen et al., 2015). The large predicted losses to aerosol influence simulation of both pRONO$_2$ (for which uptake is

the only source in the model) and HNO$_3$ (which is produced during pRONO$_2$ hydrolysis). We find here that our simulation including a large sink to aerosol is consistent with observed surface pRONO$_2$ concentrations and variability (Figs. 8 and 10), HNO$_3$ concentrations (Travis et al., 2016, Fig. 2), and nitrate wet deposition fluxes (Travis et al., 2016, Fig. 3) during SEAC$^4$RS and SOAS.

    Overall, more than 80% of simulated gas-phase RONO$_2$ are lost via processes that irreversibly remove nitrogen from

the atmosphere (deposition, aerosol hydrolysis). The remainder is primarily lost via photolysis, driven largely by the fast photolysis of 2nd generation carbonyl INs (Müller et al., 2014). RONO$_2$ lifetimes are too short (minutes-hours, Table 1) for significant transport to occur, and simulated RONO$_2$ loss typically occurs only a short distance from sources. Summed over the Southeast US domain, we find gross RONO$_2$ production and loss are roughly balanced (640 Mg N day$^{-1}$). This balance implies that BVOC-derived gas-phase RONO$_2$ are not generally exported from the Southeast US, in agreement with earlier

work (Horowitz et al., 2007; Hudman et al., 2007; Fang et al., 2010; Mao et al., 2013). However, this calculation excludes the longer-lived small alkyl nitrates and non-hydrolyzing particulate nitrates not simulated in GEOS-Chem (Sect. 4). These may be an important source of exported reactive nitrogen, and their inclusion should be a priority for future model development.

    The impacts of RONO$_2$ production and other loss processes on the NO$_x$ budget are shown in Fig. 11b for the Southeast US boundary layer in August-September 2013. Non-RONO$_2$ losses in the figure are mainly HNO$_3$ formation, with an additional

contribution from PANs (relevant in regions with elevated NO$_x$; Browne and Cohen, 2012). We find in the model that gross NO$_x$ loss due to RONO$_2$ production is 35 Gg N over this period. As shown in Fig. 11a, only 23% of this RONO$_2$ (8 Gg N) goes on to recycle NO$_x$. We therefore find that RONO$_2$ production serves as a net NO$_x$ sink of 27 Gg N in the Southeast US in summer, equivalent to 21% of NO$_x$ emitted in this region and season.





These regional-scale averages conceal important spatial variability. Figure 12 shows how the $NO_x$ sink due to $RONO_2$ production varies spatially across the Southeast US in summer, and how this depends on the ratio between BVOC and $NO_x$ emissions ($E_{BVOC}/E_{NOx}$). The fractional $NO_x$ sink to $RONO_2$ is strongly correlated ($r = 0.90$) to the $E_{BVOC}/E_{NOx}$ ratio. Our finding that $RONO_2$ production dominates $NO_x$ loss in very low $NO_x$ environments is consistent with an earlier analysis

for boreal Canada (Browne and Cohen, 2012), which found the fractional sink to $RONO_2$ approached unity for $[NO_x] < 50$ ppt, and with analysis of a subset of the SEAC[4]RS data from the low-$NO_x$ Ozarks Mountains (Wolfe et al., 2015).

Figure 12c shows how the fractional $NO_x$ sink to $RONO_2$ (blue) and the $E_{BVOC}/E_{NOx}$ emission ratio (red) vary as a function of $NO_x$ emissions (gray, shown as their cumulative distribution binned into 5% quantiles). Both are inversely related to $NO_x$ emissions. We see from the figure that $RONO_2$ production is the dominant $NO_x$ sink for regions that account for the

lowest 5% of total Southeast US $NO_x$ emissions (leftmost bar in Fig. 12c), but the importance of the sink drops off rapidly as $NO_x$ emissions increase. By the time 30% of the regional $NO_x$ emissions are accounted for, the fractional sink has dropped to 0.2, and from there continues to decline to a minimum of 0.03 in the highest-emitting regions.

The mean $E_{BVOC}/E_{NOx}$ ratio averaged over the Southeast US is 5.3 and is highlighted as the white point in Fig. 12c. The figure shows that most Southeast US $NO_x$ emission ($\sim$65%) occurs at $E_{BVOC}/E_{NOx}$ ratios that are significantly lower than

the regional mean, highlighting the significant spatial segregation between $NO_x$ and BVOC emissions in this region (Yu et al., 2016).

Emissions projections for the Southeast US anticipate continued decreases in $NO_x$ emissions (and concomitant increases in the $E_{BVOC}/E_{NOx}$ ratio). While these changes should increase the importance of $RONO_2$ for the $NO_x$ budget, the relationship shown in Fig. 12c suggests very large emissions decreases will be necessary before $RONO_2$ becomes a major regional sink

for NOx. The figure shows that the sink to $RONO_2$ is only sensitive to $NO_x$ emissions in regions where they are already low: a 10% decrease in total Southeast US $NO_x$ emissions (e.g., a leftward shift by two bars in the figure) would increase the importance of the sink by less than 0.5%. The actual rate at which $NO_x$ emissions in the Southeast US will decrease varies widely among different projections. Under the Representative Concentration Pathway 8.5 (RCP8.5), for example, the Southeast US would see a decrease (relative to 2013 emissions) of 45% by 2050 to $\sim$1300 Mg N day$^{-1}$; according to Fig. 12, the $RONO_2$

sink would still only account for about 10% of the loss in the highest emitting regions. Under the more aggressive RCP4.5, emissions would decline by 65% to $\sim$800 Mg N day$^{-1}$ in 2050. At this stage, the $RONO_2$ sink would become significant (>20%) throughout the region.

# 6  Conclusions

We have used airborne and ground-based observations from two summer 2013 campaigns in the Southeast US (SEAC[4]RS,

SOAS) to better understand the chemistry and impacts of alkyl and multi-functional organic nitrates ($RONO_2$). We used the observations, along with findings from recent laboratory, field, and modeling studies, to update and evaluate biogenic volatile organic compound (BVOC) oxidation schemes in the GEOS-Chem chemical transport model (CTM). From there, we





used the updated CTM with $0.25° \times 0.3125°$ ($\sim 25 \times 25$ km$^2$) horizontal resolution to examine RONO$_2$ speciation, chemical production/loss processes, and importance as a sink for NO$_x$.

Our improved mechanism provides a state-of-the-science description of isoprene oxidation in the presence of NO$_x$, with updates including a 9% isoprene nitrate (IN) yield (Xiong et al., 2015), an increase in the population of $\beta$- vs $\delta$-hydroxyl isomers (Peeters et al., 2014), revised IN reaction rate constants and products (Jacobs et al., 2014; Lee et al., 2014), fast photolysis of carbonyl INs (Müller et al., 2014), rapid IN dry deposition (Nguyen et al., 2015), and a simplified scheme for aerosol partitioning of soluble INs (Xu et al., 2014; Marais et al., 2016) followed by particle-phase hydrolysis (Jacobs et al., 2014; Rindelaub et al., 2015). For the first time in GEOS-Chem, we have also added both OH- and NO$_3$-initiated monoterpene oxidation leading to the formation of monoterpene nitrates (MTNs), with similar loss processes as for INs. With these updates, GEOS-Chem simulates surface-level BVOC and RONO$_2$ mixing ratios that are generally within the observed variability of the SEAC$^4$RS and SOAS data.

Observed first generation IN (ISOPN) variability is generally reproduced without bias by GEOS-Chem, except at midday when modeled ISOPN peaks while SOAS observations indicate a gradual decline. For second generation INs, the model shows more skill for species produced primarily from $\beta$–hydroxyl isomers (MVKN+MACRN) than those from $\delta$-hydroxyl isomers and NO$_3$-initiated chemistry (PROPNN+ETHLN). For the latter, GEOS-Chem underestimates both magnitudes and variability relative to the SEAC$^4$RS observations. While this could imply a more important role for $\delta$-channel oxidation than included in our mechanism, theoretical considerations suggest that our assumed $\delta$-hydroxyl contribution is already an upper limit (Peeters et al., 2014), and more measurements are needed to reconcile these theoretical and observational constraints. Better understanding of nighttime NO$_3$-initiated isoprene oxidation could also play an important role in improving simulation of second generation INs.

The SEAC$^4$RS observations imply that gas-phase INs account for 25-50% of total surface RONO$_2$, much less than inferred from previous modeling studies (Mao et al., 2013; Xie et al., 2013). GEOS-Chem reproduces this contribution and attributes an additional 10% of RONO$_2$ to MTNs. Both observations and model show 10-20% of the remaining RONO$_2$ at the surface is in the particle phase (pRONO$_2$). In the free troposphere, GEOS-Chem greatly underestimates total RONO$_2$ by ignoring contributions from small, long-lived nitrates derived from anthropogenic VOCs and from non-hydrolyzing particulate species. This has a significant impact on simulation of reactive nitrogen export from the United States and should be remedied in future model development.

We find in the model that formation of pRONO$_2$ via aerosol uptake, followed by particle-phase hydrolysis, is the dominant loss process for gas-phase RONO$_2$. Including this large sink to aerosol results in simulated RONO$_2$, pRONO$_2$ and HNO$_3$ mixing ratios and nitrate deposition fluxes that are consistent with observations. RONO$_2$ loss via deposition is also significant, with RONO$_2$ (both gas-phase and particulate) responsible for $\sim$3% of total N deposition over the Southeast US in summer.

Overall, less than a quarter of simulated gas-phase RONO$_2$ loss recycles atmospheric NO$_x$. We find in the model that RONO$_2$ production accounts for 21% of the net sink of NO$_x$ emitted in the Southeast US in summer. RONO$_2$ production is the dominant NO$_x$ sink only in regions where elevated BVOC emissions are paired with very low NO$_x$ emissions. Elsewhere,





the importance of the sink declines rapidly as a function of $NO_x$ emissions. Most of the Southeast US $NO_x$ is emitted in locations where BVOC emissions are relatively low, limiting the importance of $RONO_2$ as a $NO_x$ sink.

Southeast US $NO_x$ emissions have been declining for the past two decades (Hidy et al., 2014; Simon et al., 2015) and further reductions are projected (Lamarque et al., 2011; EPA, 2014). Previous studies have suggested these declines will trigger a more

important role for $RONO_2$ as a $NO_x$ sink in future (Browne and Cohen, 2012). In contrast, we find here that the $NO_x$ sink to $RONO_2$ is only sensitive to $NO_x$ emissions in regions where they are already low because of the spatial segregation between $NO_x$ and BVOC emissions. We find that a 10% decrease in Southeast US $NO_x$ emissions would enhance the importance of this sink by less than 0.5%. $HNO_3$ formation and deposition is likely to remain the dominant sink for $NO_x$ even as $NO_x$ emissions decrease.

*Acknowledgements.* We are grateful to the entire NASA SEAC[4]RS team for their help in the field, and we thank Eleanor Browne and Fabien Paulot for helpful discussions about the monoterpene nitrate scheme. This work was funded by a University of Wollongong Vice Chancellor's Postdoctoral Fellowship to JAF and by the NASA Tropospheric Chemistry Program. This research was undertaken with the assistance of resources provided at the NCI National Facility systems at the Australian National University through the National Computational Merit Allocation Scheme supported by the Australian Government. JLJ, PCJ,and DAD were supported by NASA NNX15AH33A and NNX15AT96G.

Isoprene and monoterpene measurements during SEAC[4]RS were supported by the Austrian Federal Ministry for Transport, Innovation and Technology (bmvit) through the Austrian Space Applications Programme (ASAP) of the Austrian Research Promotion Agency (FFG). AW and TM received support from the Visiting Scientist Program at the National Institute of Aerospace (NIA).



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





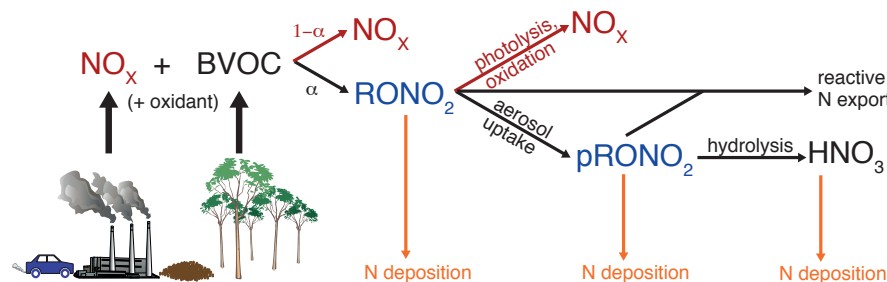

**Figure 1.** Schematic representation of organic nitrate chemistry and impacts. Organic nitrates are shown in blue, $NO_x$ and processes that recycle $NO_x$ are shown in red, and nitrogen deposition is shown in orange. Symbols courtesy of the Integration and Application Network, University of Maryland Center for Environmental Science (ian.umces.edu/symbols/).





**Figure 2.** Schematic of the formation of isoprene nitrates (INs) from OH-initiated isoprene oxidation as implemented in GEOS-Chem. The isomers shown are indicative as the mechanism does not distinguish between isomers (except for $\beta$- vs. $\delta$-configurations). For ISOPNO$_2$ oxidation, only IN products are shown, along with their yields from both NO and HO$_2$ pathways. Small yields (<10%) of MVKN and MACRN from $\delta$-ISOPNO$_2$ are not shown.





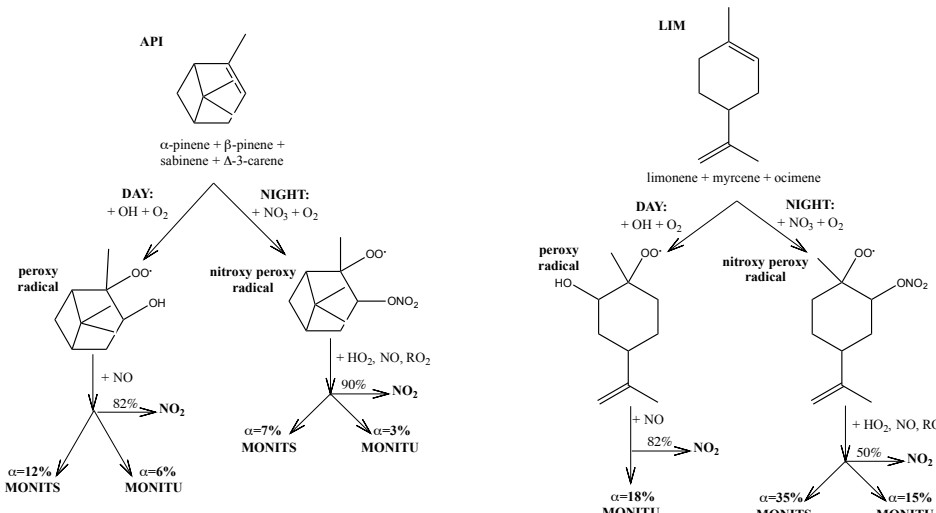

**Figure 3.** Simplified representation of the formation of monoterpene nitrates (MTN) from monoterpene oxidation as implemented in GEOS-Chem. For each lumped species, only one indicative form is shown.





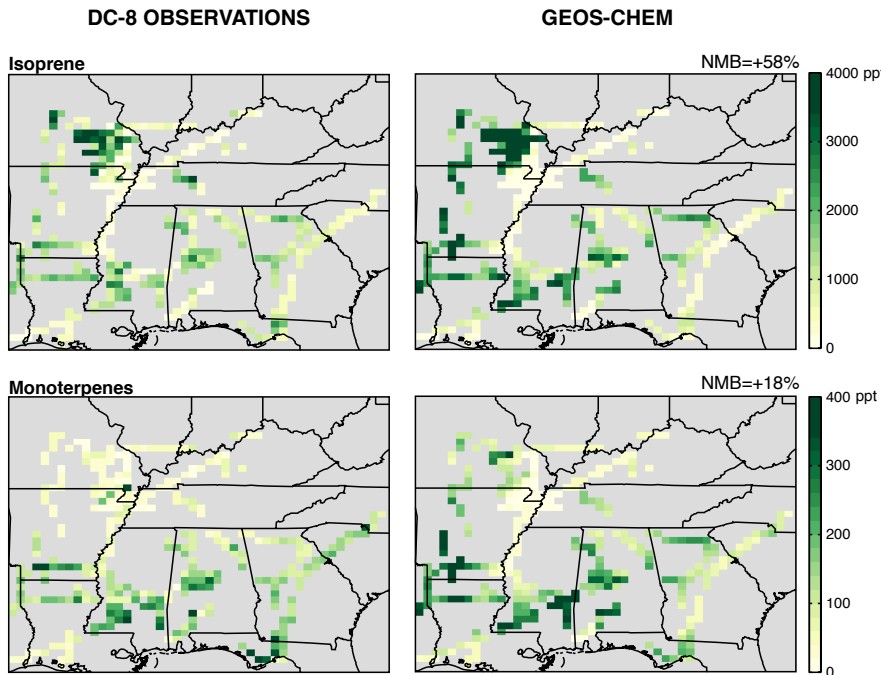

**Figure 4.** Observed (left) and simulated (right) mixing ratios of isoprene and monoterpenes below 1 km during the SEAC[4]RS aircraft campaign (12 Aug - 23 Sep 2013), along with the normalized mean bias (NMB) of the simulation relative to the PTR-MS measurements. The GEOS-Chem model has been sampled along the aircraft flight tracks, and the observations binned to the spatial and temporal resolution of the model.





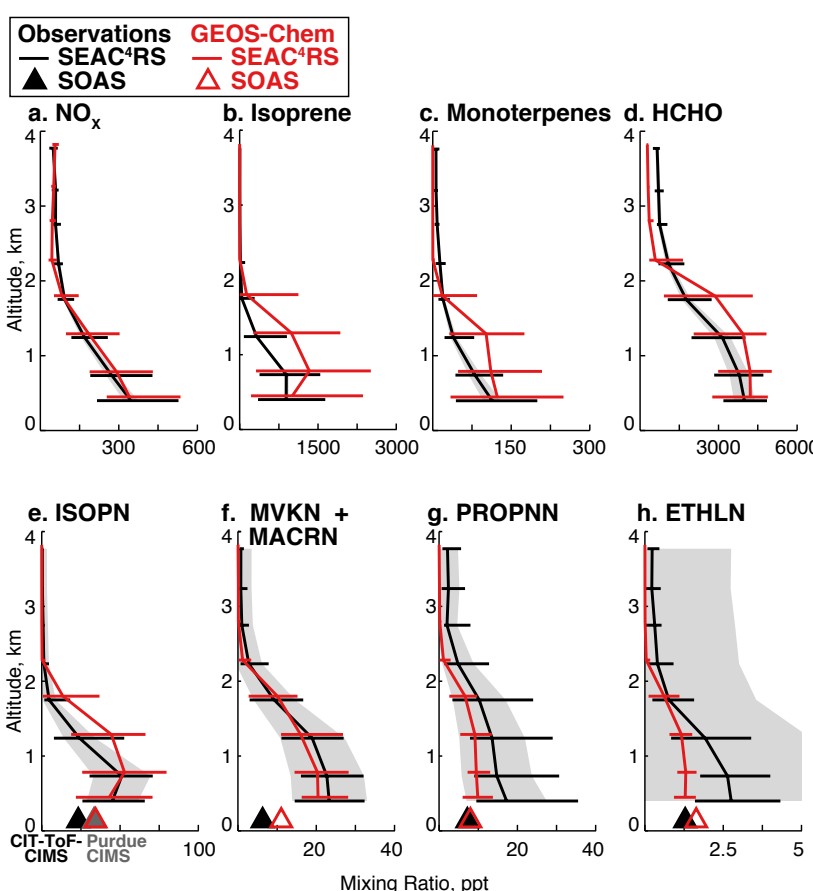

**Figure 5.** Observed (black) and modeled (red) median 0-4 km profiles of $NO_x$, biogenic VOCs, and oxidation products over the Southeast US (80-94.5°W, 29.5-40°N) during $SEAC^4RS$. Data are binned in 500m increments, and horizontal lines indicate the interquartile range within each bin. Gray shading represents the measurement uncertainty. The model has been sampled in the same manner as the observations, as described in the text. For organic nitrates (e-h), SOAS campaign median surface values are shown as triangles. For ISOPN (e), the gray triangle represents the Purdue CIMS and the black triangle the CIT-ToF-CIMS.





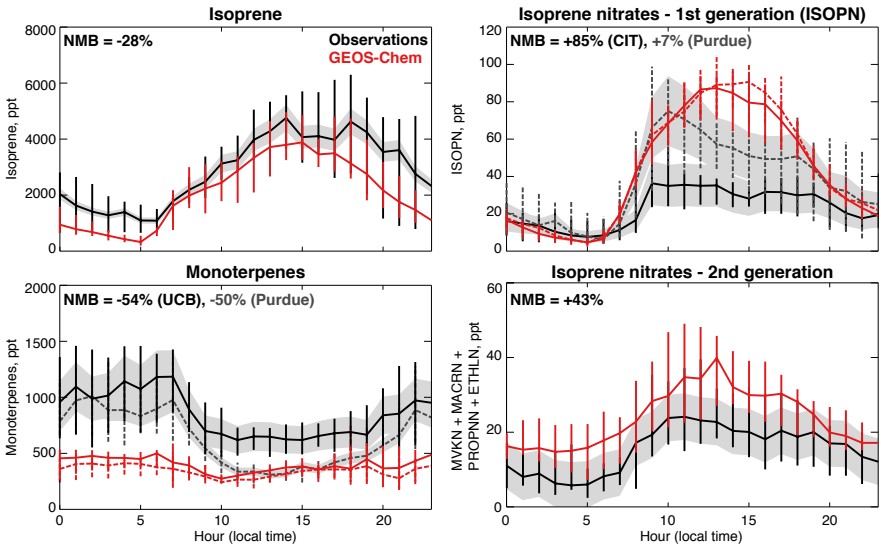

**Figure 6.** Observed (black, gray) and simulated (red) median diurnal cycles of isoprene, monoterpenes, first generation isoprene nitrates (ISOPN), and second generation isoprene nitrates (MVKN+MACRN+PROPNN+ETHLN) at Centreville during the 2013 SOAS campaign. Gray shading represents the measurement uncertainty, vertical bars show the interquartile range of the hourly data, and the normalized mean bias (NMB) of the simulation is given inset. The model has been sampled in the Centreville grid box only for hours with available data during 16 June - 11 July for isoprene and monoterpenes from the UC Berkeley PTR-ToF-MS (solid black), 13 June - 15 July for $\alpha$-pinene + $\beta$-pinene from the Purdue 2D-GF-FID (dashed gray), 1 June - 11 July for ISOPN from the Purdue CIMS (dashed gray), and 1 June - 4 July for ISOPN and 2nd generation isoprene nitrates from the CIT-ToF-CIMS (solid black). For ISOPN and monoterpenes, differences in data availability between the two measurements result in slightly different model values (solid/dashed red lines).

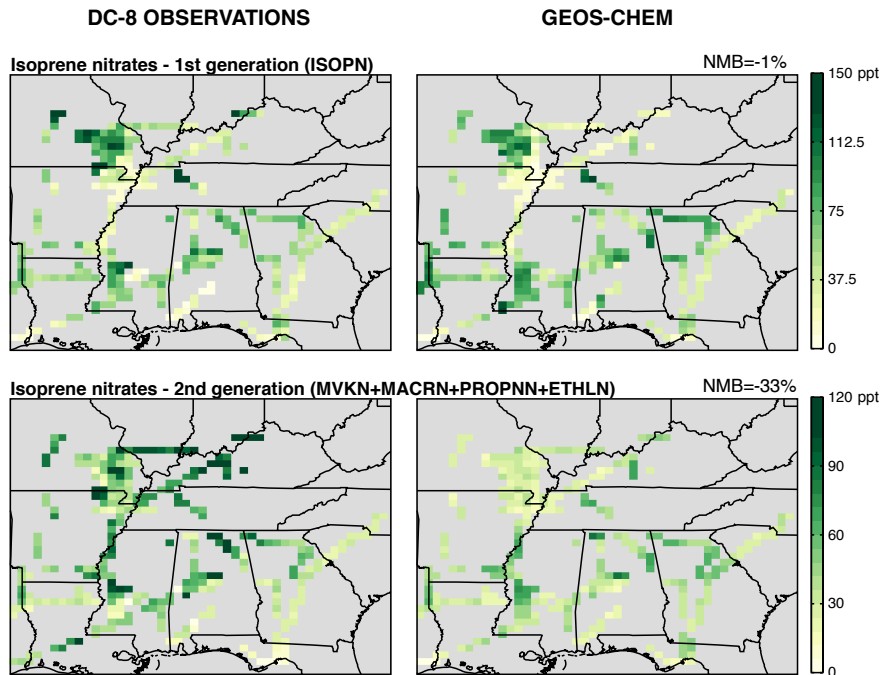

**Figure 7.** Observed (left) and simulated (right) mixing ratios of isoprene nitrates below 1 km during SEAC$^4$RS, separated into first generation (ISOPN) and second generation (MVKN+MACRN+PROPNN+ETHLN) species, along with the normalized mean bias (NMB) of the simulation. The GEOS-Chem simulation has been sampled along the aircraft flight tracks, and the observations binned to the spatial and temporal resolution of the model, as described in the text.





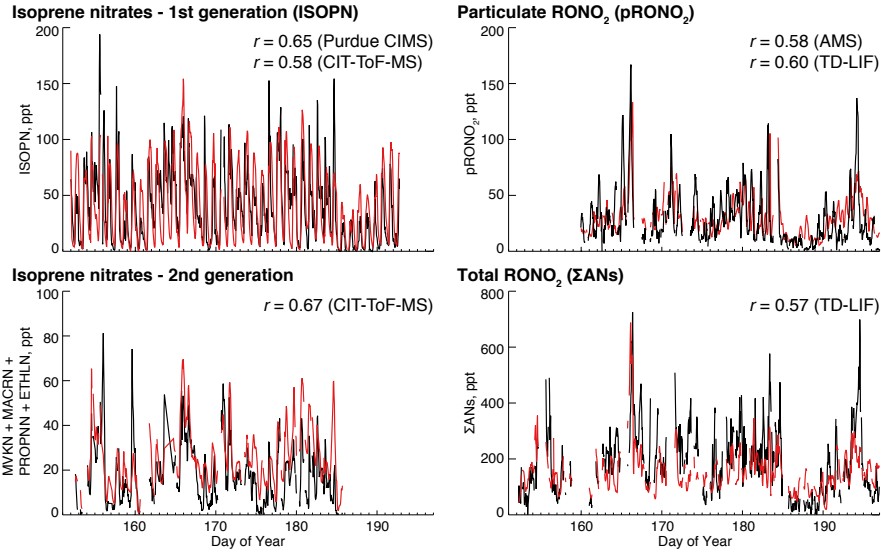

**Figure 8.** Timeseries of observed (black) and simulated (red) hourly mean $RONO_2$ at Centreville during the 2013 SOAS campaign for 1st generation isoprene nitrates (ISOPN, from the Purdue CIMS), 2nd generation isoprene nitrates (MVKN+MACRN+PROPNN+ETHLN from the CIT-ToF-CIMS), particulate $RONO_2$ (pRONO$_2$, from the AMS) and total alkyl nitrates ($\Sigma$ANs, from the TD-LIF). The model has been sampled in the Centreville grid box only for hours with available data from each instrument. The model-observation correlation coefficient ($r$) for each species is given inset both for the measurement shown and (where available) for additional measurement of the same species (with timeseries shown in Fig. S4).

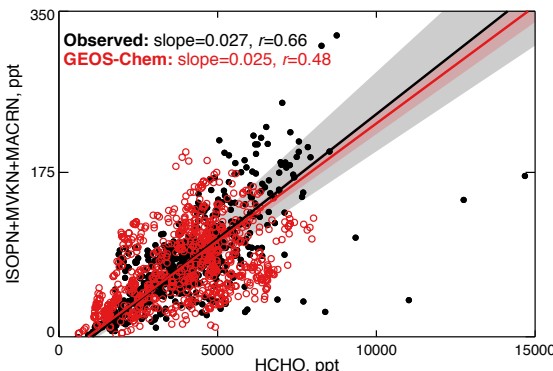

**Figure 9.** Observed (black) and simulated (red) correlations between HCHO and the sum of major isoprene nitrates produced via daytime isoprene oxidation (ISOPN+MVKN+MACRN) in Southeast US surface air (<1 km) during SEAC[4]RS. Thick solid lines indicate the best fit as calculated from a reduced major axis regression, and shaded areas show the 95% confidence interval on the regression slope as determined by bootstrap resampling. The regression slopes and correlation coefficients are given inset.



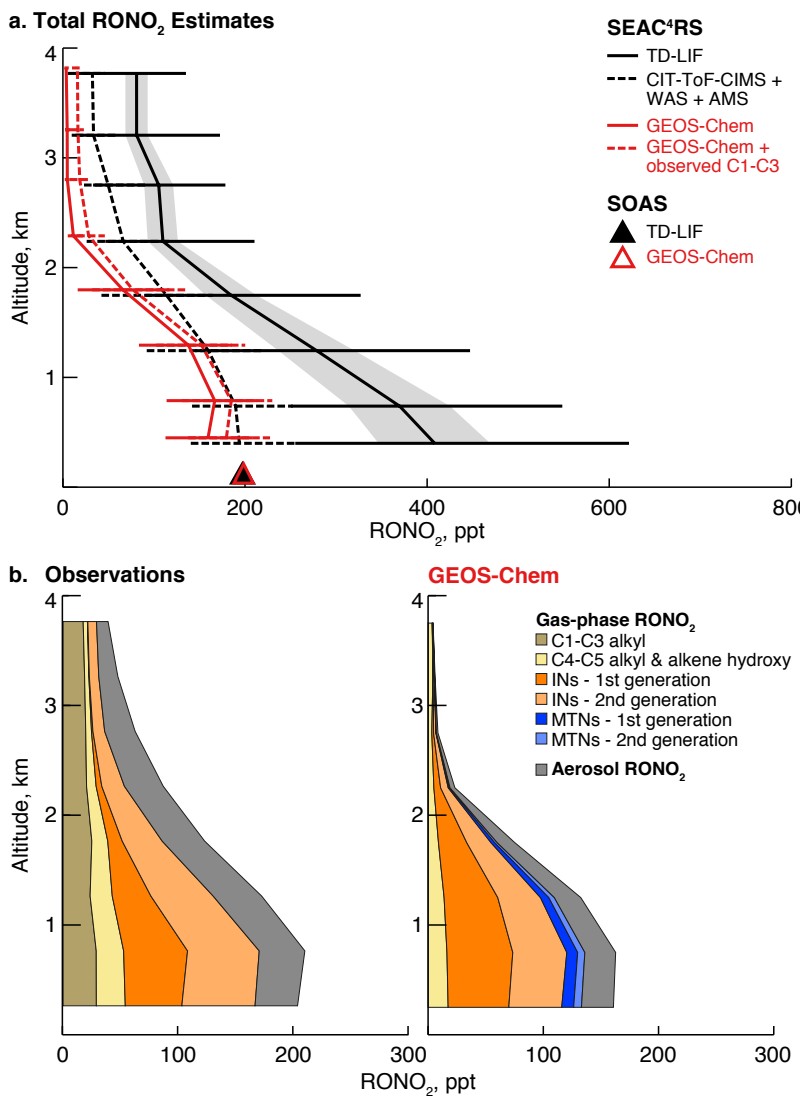

**Figure 10.** (a) Median vertical profiles of estimated total $RONO_2$ over the Southeast US during $SEAC^4RS$. For the observations, the solid black line indicates the TD-LIF $\Sigma$ANs measurements (with gray shading for the measurement uncertainty) and the dashed black line the sum of CIT-ToF-CIMS, WAS, and AMS measurements of individual $RONO_2$ species (gas-phase and particulate). For the model, the solid red line indicates the total simulated $RONO_2$ and the dashed red line the sum of total simulated $RONO_2$ plus measured $\leq C_3$ RONO2 that are not included in the simulation. Triangles compare the total $RONO_2$ during SOAS from TD-LIF $\Sigma$ANs and GEOS-Chem. (b) Mean $RONO_2$ composition from the observations (CIT-ToF-CIMS, WAS, and AMS) and the model. Isoprene nitrates (INs) include 1st generation (ISOPN, plus ISN1 for GEOS-Chem) and 2nd generation INs (MVKN+MACRN, PROPNN, ETHLN, NISOPOOH, plus DHDN for GEOS-Chem). Monoterpene nitrates (MTNs) are shown for the model only and include first and second generation contributions. Other gas-phase $RONO_2$ (yellow, brown) are mainly anthropogenic and do not represent the same species between the model and the observations.





**a. Gas-phase RONO$_2$ Sinks (642 Mg N day$^{-1}$)**

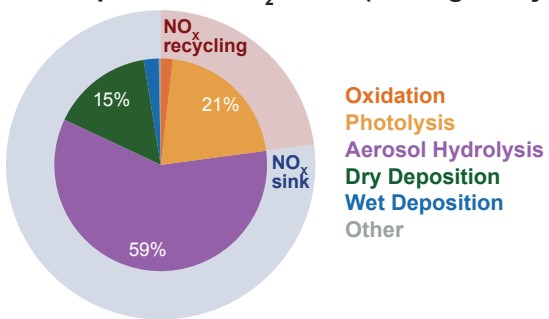

**b. Terminal NO$_x$ Sinks (2.18 Gg N day$^{-1}$)**

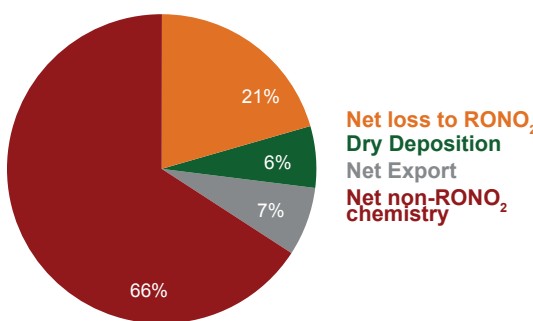

**Figure 11.** Simulated relative importance of gas-phase loss processes (%) in the Southeast US boundary layer (80-94.5°W, 29.5-40°N, <2 km) during August-September 2013 for (a) gas-phase RONO$_2$ and (b) NO$_x$. In (a), outer circles group losses into those that recycle NO$_x$ (pale red) and those that serve as terminal NO$_x$ sinks (pale blue). Loss processes that recycle RONO$_2$ by converting between RONO$_2$ species (e.g. first to second generation) are not included. In (b), net loss to RONO$_2$ is calculated as the difference between NO$_x$ consumed during RONO$_2$ production and NO$_x$ recycled during RONO$_2$ loss, with recycling efficiencies from (a). Net non-RONO$_2$ chemistry is the difference between NO$_x$ chemical production and chemical loss excluding all RONO$_2$ chemistry, and net export is the difference between emissions and all other sinks. Absolute loss rates from all processes combined (Mg N day$^{-1}$) are given in the sub-plot titles.





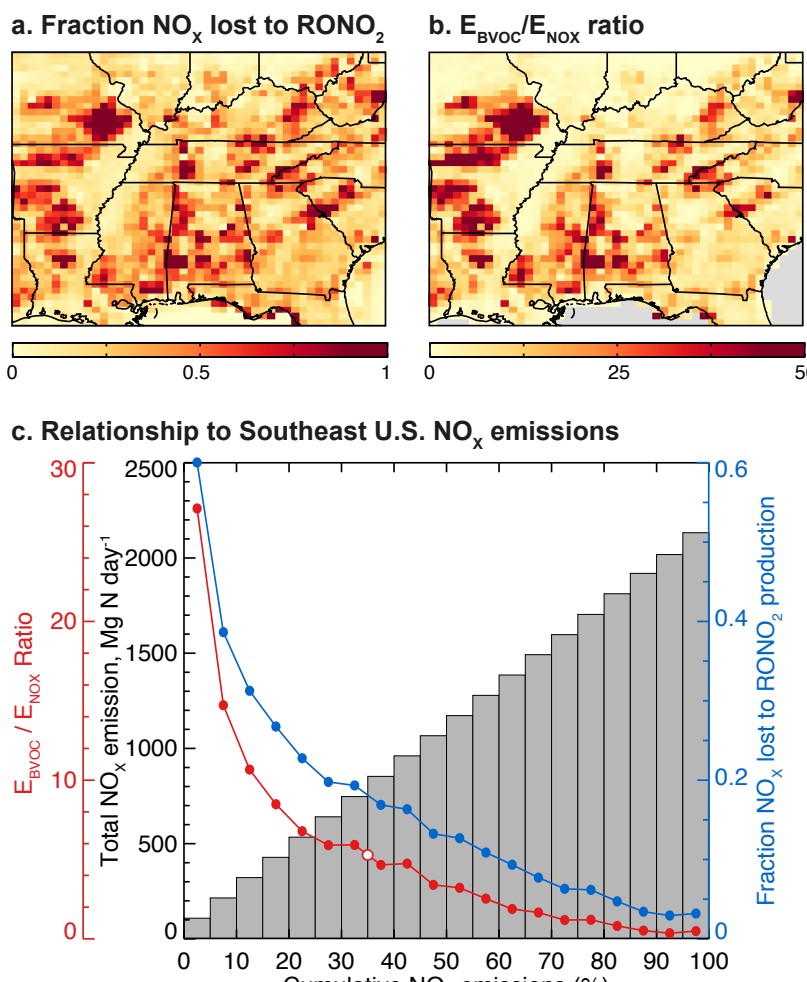

**Figure 12.** Importance of organic nitrates as a sink for $NO_x$, as a function of BVOC and $NO_x$ emissions. (a) Simulated fraction of emitted $NO_x$ that is lost to $RONO_2$ production in Southeast US surface air. (b) Ratio of BVOC (isoprene + monoterpene) emissions ($E_{BVOC}$) to $NO_x$ emissions ($E_{NOx}$). (c) Mean values of variables from (a) and (b) as a function of cumulative $NO_x$ emissions in the Southeast US. Model grid squares have been sorted by $NO_x$ emissions then grouped into bins that each represent 5% of total Southeast US $NO_x$ emissions. Values shown for the fractional $NO_x$ sink due to $RONO_2$ production (blue) and the mean $E_{BVOC}/E_{NOx}$ emissions ratio (red) represent the mean within each bin. $NO_x$ emissions are shown as cumulative totals (gray) in Mg N day$^{-1}$. The Southeast US mean $E_{BVOC}/E_{NOx}$ emissions ratio (5.3) is highlighted with a white circle.





**Table 1.** Gas-phase organic nitrates in GEOS-Chem along with their principal formation pathways, removal processes (ordered by importance), and lifetimes in Southeast US surface air during SEAC[4]RS.

| Species | Model name | Principal formation pathways[a] | Removal Processes[b] | Lifetime (hr)[c] |
|---|---|---|---|---|
| $\beta$-hydroxy isoprene nitrate | ISOPNB | ISOP + OH | aerosol hydrolysis deposition oxidation photolysis | 1.8 |
| $\delta$-hydroxy isoprene nitrate | ISOPND | ISOP + OH | deposition oxidation aerosol hydrolysis photolysis | 4.0 |
| $C_5$ nitrooxy carbonyl | ISN1 | ISOP + $NO_3$ | deposition photolysis oxidation aerosol hydrolysis | 0.29 |
| $C_5$ nitrooxy hydroperoxide[d] | INPN | ISOP + $NO_3$ | n/a | n/a |
| methyl vinyl ketone nitrate | MVKN | ISOPNB + OH | deposition aerosol hydrolysis photolysis oxidation | 3.1 |
| methacrolein nitrate | MACRN | ISOPNB + OH | photolysis deposition aerosol hydrolysis oxidation | 1.5 |
| propanone nitrate | PROPNN | ISOPND + OH ISN1 + $NO_3$ | deposition photolysis oxidation | 3.3 |
| ethanal nitrate | ETHLN | ISOPND + OH | deposition photolysis oxidation | 1.5 |
| $C_5$ dihydroxy dinitrate | DHDN | ISOPND + OH ISOPND + OH | aerosol hydrolysis deposition | 4.6 |
| saturated first generation monoterpene nitrate | MONITS | API + OH API + $NO_3$ LIM + $NO_3$ | deposition aerosol hydrolysis oxidation photolysis | 1.8 |
| unsaturated first generation monoterpene nitrate | MONITU | API + OH API + $NO_3$ LIM + OH LIM + $NO_3$ | oxidation deposition aerosol hydrolysis photolysis | 0.85 |
| second generation monoterpene nitrate | HONIT | MONITU + OH MONITS + OH | aerosol hydrolysis deposition photolysis oxidation | 1.7 |

[a] Primary precursor(s) and associated oxidant(s). The related peroxy radicals and their oxidants can be seen in Figs. 1 and 2.

[b] Removal processes for each species are ordered by their contribution to total loss during SEAC[4]RS. Losses due to oxidation, photolysis, and aerosol uptake are calculated along the SEAC[4]RS flight tracks. Deposition includes both dry and wet scavenging and is calculated from regional means over all Southeast US grid boxes. Wet deposition in the model is calculated for lumped species ISOPNB+ISOPND, MVKN+MACRN, and MONITU+MONITS and individually for all others. For this table, we assume partitioning of 90% ISOPNB (10% ISOPND) based on the initial formation yields and a 50:50 split for the other lumped species. Wet scavenging is only a small contribution to total $RONO_2$ deposition, and this assumption has minimal impact on these values.

[c] Lifetimes are the combined lifetimes against deposition as calculated over all grid boxes and against oxidation, photolysis, and aerosol hydrolysis as calculated along the flight tracks, with further details in note $b$.

[d] INPN is not treated as a transported species, so diagnostics needed to calculate removal rates and lifetime are not available.