# Peer review of "Organic nitrate chemistry and its implications for nitrogen budgets in an isoprene- and monoterpene-rich atmosphere: constraints from aircraft (SEAC4RS) and ground-based (SOAS) observations in the Southeast US"

_Atmospheric Chemistry and Physics, 2016_

## Referee Comment (RC1) · Anonymous Referee #2 · 2 Mar 2016

Hydrolysis is receiving increased attention in chemical transport models as a way to remove organic nitrate (thus NOx) from the atmosphere by converting it to nitric acid. Hydrolysis that has been implemented in (CTMs) chemical transport models thus far (CAMx: Hildebrant Ruiz and Yarwood 2013, CMAQ: Pye et al. 2015) assumes that hydrolysis occurs after semivolatile organics partition to an organic-rich particle. The process of semivolatile organics (monoterpene-derived) undergoing hydrolysis has been demonstrated in laboratory work (e.g. Boyd et al. 2015, Bean and Hildebrandt Ruiz

2016). The timescales for hydrolysis used in CTMs ranges from 3 to 6 hours or longer. Thus, the work of Fisher et al., with a 1 hour hydrolysis represents a significantly faster rate of hydrolysis. Indeed, more than half of the organic nitrates in table 1 have a sub 2-hour lifetime. The irreversible reactive uptake formulation used by Fisher et al. also differs from the equilibrium vapor-pressure based approach. As a result, the Fisher et al. work represents a significantly different approach than has been used thus far for hydrolysis and creation of particulate organic nitrate. The implications for ozone and aerosol may motivate future work to adopt similar formulations and parameter choices should be fully justified.

My main question is whether or not another set of parameters than those examined here would provide equally good model performance. For example, there are no sensitivity simulations to indicate if the 1 hour hydrolysis is significantly better than say a 6 hour hydrolysis. It would be useful to know if another set of conditions (specifically lower BVOC emissions, higher NOx emissions, slower hydrolysis, slower uptake of organic nitrates) also leads to a consistent picture of southeast chemistry. More detailed comments follow.

Travis et al. is referenced as the basis for the gas-phase oxidant chemistry in this work. In that work, they indicate NOx emissions from the EPA NEI are too high by 50%. I assume the simulations in this work (Fisher et al.) reduce NOx emissions consistent with Travis et al. Given the suspected importance of reduced NOx emissions, I provide some comments on the Travis et al. paper (available at: http://acmg.seas.harvard.edu/publications/2016/Travis_ACPD_2016.pdf). Reducing the NEI NOx by ∼50% may imply a larger hydrolysis sink is needed in the model than is actually necessary. Comments on the Travis et al. paper:

1. Given the missing free tropospheric NO2 in GEOS-Chem and the fact that boundary layer NO2 is less than 30% of the column, how quantitative can the NO2 column evaluation be? What is the error associated with the BEHR and NASA columns? It seems like the GEOS-Chem shape factors may not be a good representation of the

NO2 profile given the lack of NO2 aloft. What is the detection limit/precision for surface NO2 from OMI?

2. The ratio of ISOPOOH to ISOPN seems degraded with the NOx emission reduction compared to the base indicating too much RO2+HO2 vs RO2+NO. Are there additional later generation products that can be examined to justify proper branching?

3. How does the magnitude of the soil NOx emission predicted in GEOS-Chem compare to other estimates? Does the NEI provide a comparable estimate?

Fisher et al. (this work) comments:

4. Figure S3 indicates too many pRONO2 compared to the AMS data during the day, particularly 13-17 local time. The diurnal profile of the model pRONO2 (relatively flat with peak around 15 local time) also doesn't match the observed AMS profile (peak at night around 3-4 am with minimum around 15 local time). Are the isoprene nitrates too aggressively put in the particle?

5. I recommend performing a total NOy and total inorganic nitrate (HNO3 + NO3-) evaluation. I didn't see a total NOy evaluation in either Travis et al. or Fisher et al. It was a little unclear if Travis et al. included all the updates (ie monoterpenes) of Fisher et al. In any case, Travis shows that HNO3 is high in the base. If the base includes hydrolysis, that may explain why. Furthermore, total RONO2 is underestimated by ∼50% according to Figure 10. This implies there are missing NOx sinks (or sources) in the model.

6. Figure 12 shows that as the ratio of BVOC to NOx emissions increases, the fraction of NOx lost to RONO2 (vs HNO3) increases. If BVOC emissions are biased high (Fig 4 shows a NMB of 58% for isoprene and 18% for monoterpenes compared to DC8 flights) and NOx emissions are biased low (2011 NEI is reduced ∼50%) then the model will predict too much NOx lost to RONO2 vs HNO3. Hydrolysis converts RONO2 to HNO3. Another set of BVOC and NOx emission levels and their evaluation would be useful.

It would be nice to see analysis examining the possibility of lower BVOC emissions, higher NOx emissions, slower hydrolysis, and slower uptake of organic nitrates leading to a consistent picture of southeast chemistry.

7. Page 7, line 32: The yield of organic nitrates from API+NO3 is set to 10%. This seems to be on the low end considering the values tabulated in Fry et al. 2014 which range from 11-29% for a-pinene and 40-74% for b-pinene. Is there a specific RO2 fate assumed for these yields?

8. Page 8, line 33: Is there a figure that can be referenced to show good RONO2 performance?

9. Page 9, near line 7, how was the specific lifetime of 1 hour against hydrolysis chosen? Were other values used in preliminary simulations? For the isoprene system, what is the fraction of tertiary vs. non-tertiary nitrates?

10. Page 9, line 16, in reality, the hydrolysis lifetime should affect gas-phase RONO2. In the reactive uptake formulation of Marais et al. the uptake coefficient depends on the rate of particle-phase reaction (here hydrolysis). Faster reaction increases the uptake coefficient. It should be pointed out that this dependence of gamma on hydrolysis is not captured by the model.

11. Page 13: Are the TD-LIF measurements of total ANs consistent with the SEARCH network measurements during SOAS?

12. A few things indicate the NOx emissions may have been reduced too much:

o the total ANs are low by ∼50% (Figure 10)

o The ratio of ISOPOOH to ISOPN seems degraded (too high) with the NOx emission reduction compared to the base (Travis et al. Figure 2)

o The NO2 columns are low compared to observations (Travis et al.) and the ability of the satellite products to determine the magnitude of a given error in surface NO2 is

unclear

o A source of HNO3 (hydrolysis) must be added to the model

13. In addition to posting the mechanism online, can full details be provided in the supporting material for future reference?

14. While I am aware of studies of isoprene nitrates undergoing hydrolysis in bulk systems (e.g. Jacobs et al. 2014 ACP), what studies explore the interaction of gas-phase isoprene nitrates and aerosol-phase hydrolysis products for the isoprene system?

References

Bean, J. K. and Hildebrandt Ruiz, L.: Gas–particle partitioning and hydrolysis of organic nitrates formed from the oxidation of $\alpha$-pinene in environmental chamber experiments, Atmos. Chem. Phys., 16, 2175-2184, doi:10.5194/acp-16-2175-2016, 2016.

Hildebrandt Ruiz, L.; Yarwood, G. Interactions between organic aerosol and NOy: influence on oxidant production, AQRP Project 12–012; Report, 2013, aqrp.ceer.utexas.edu/projectinfoFY12_13/12-012/12-012%20Final%20Report.pdf.

Jacobs, M. I., Burke, W. J., and Elrod, M. J.: Kinetics of the reactions of isoprene-derived hydroxynitrates: gas phase epoxide formation and solution phase hydrolysis, Atmos. Chem. Phys., 14, 8933-8946, doi:10.5194/acp-14-8933-2014, 2014.

---

## Referee Comment (RC2) · Anonymous Referee #1 · 8 Mar 2016

This paper advances our understanding of isoprene and monoterpene degradation chemistry in a regional modelling framework. It specifically implements a range of updates to the isoprene and monoterpenes derived nitrates and compares the model results to those observed by field campaigns in the southern US. These observational datasets provide a uniquely comprehensive assessment of the species produced through this chemistry. The paper shows the role of a range of new processes notably the heterogeneous processing of RONO2 species.

This paper is comprehensive, well written and advances our understanding of this important area of atmospheric chemistry. My suggestion is that the paper is published but have one potential addition to the paper and a couple of minor comments below.

My major difficulty reading the paper is that there is little discussion of the impact of the addition of all of this chemistry on the wider composition of the atmosphere. How different are the North American concentrations of NOx, O3, PM, OH etc after all of this new chemistry has been added? How much difference does this make overall to the chemistry of the atmosphere? Should this chemistry be included in all modelling or is it of niche interest? This final summing up seems to be missing from the paper which makes it a little dissatisfying overall. The authors presumably have simulations with the old chemistry and the new chemistry or could readily perform a calculation with the isoprene / monoterpene nitrate chemistry switched off etc and a new section which provides some details of the overall impact would give the reader some sense of how important this is and whether they should care or not about the chemical detail described in the paper.

The authors gloss over a few of their mechanistic choices. They should put the chemical mechanism used in the work into supplementary material for both the gas, heterogeneous and aerosol phase chemistry and include more details of how they have changed absorption cross-sections from the IUPAC recommendation (presumably the quantum yield is 1?).

In a couple of places we are re-assured that the kinetic choices provide the best fit to the observational data ("Although simplified, we find this parameterization improves the model fit relative to the SEAC4RS and SOAS …..", "In any case, the choice of hydrolysis lifetime does not affect the concentration of gas-phase RONO2 species (because pRONO2 cannot re-partition to the gas phase in the model), and we find this value provides a reasonable match to AMS measurements of total pRONO2 at the surface during SOAS and SEAC4RS"). Could these results in the supplementary material so that the readers can judge these for themselves.

I found sections 3,4,5 rather long for my taste. Would it be possible to put in some sub-section headings to help split those up and allow the reader to find specific bits of information more easily?

---

## Author Comment (AC1) · 13 Apr 2016

We thank the two reviewers for their careful reading of the manuscript and their detailed comments. Our responses are shown below in green, with quoted text indented and new text in **bold.**

**Anonymous Referee #1**

This paper advances our understanding of isoprene and monoterpene degradation chemistry in a regional modelling framework. It specifically implements a range of updates to the isoprene and monoterpenes derived nitrates and compares the model results to those observed by field campaigns in the southern US. These observational datasets provide a uniquely comprehensive assessment of the species produced through this chemistry. The paper shows the role of a range of new processes notably the heterogeneous processing of RONO2 species.

This paper is comprehensive, well written and advances our understanding of this important area of atmospheric chemistry. My suggestion is that the paper is published but have one potential addition to the paper and a couple of minor comments below.

My major difficulty reading the paper is that there is little discussion of the impact of the addition of all of this chemistry on the wider composition of the atmosphere. How different are the North American concentrations of NOx, O3, PM, OH etc after all of this new chemistry has been added? How much difference does this make overall to the chemistry of the atmosphere? Should this chemistry be included in all modelling or is it of niche interest? This final summing up seems to be missing from the paper which makes it a little dissatisfying overall. The authors presumably have simulations with the old chemistry and the new chemistry or could readily perform a calculation with the isoprene / monoterpene nitrate chemistry switched off etc and a new section which provides some details of the overall impact would give the reader some sense of how important this is and whether they should care or not about the chemical detail described in the paper.

We agree with the reviewer that quantifying the impacts of the chemistry changes on the wider atmospheric composition would be of great interest – and this will be the topic of a follow-up paper. We would like to separate this analysis from the current paper because it is already very long and is thematically focused on the organic nitrates, so introducing additional analysis of $NO_x$, $O_3$, OH, etc. would weaken the overall focus, increase the length, and decrease the readability.

In addition, testing the old vs. new chemistry will require significant additional computational burden, as our model development tests were all performed at coarse resolution and each change was added linearly. A thorough accounting of the impacts will require multiple additional simulations, ideally with each change tested independently and at a finer resolution than used for our test simulations. We prefer to leave this more systematic analysis to a subsequent paper.

The authors gloss over a few of their mechanistic choices. They should put the chemical mechanism used in the work into supplementary material for both the gas, heterogeneous and aerosol phase chemistry and include more details of how they have changed absorption cross-sections from the IUPAC recommendation (presumably the quantum yield is 1?).

Updates to the isoprene oxidation mechanism are provided in the Supplement to Travis et al. (2016), and we now state this in the text at the start of Section 2.1:

> **All updates to the isoprene oxidation mechanism are provided in Travis et al. (2016) Tables S1 and S2.**

We now provide the monoterpene nitrate scheme in the Supplement, and state this in the text at the start of Section 2.2:

> Our implementation is summarized in Fig. 3 and described briefly below, with the full mechanism available **in the Supplement (Tables S1-S3) and** at http://wiki.seas.harvard.edu/geos-chem/index.php/Monoterpene_nitrate_scheme.

We now provide the aerosol uptake coefficients in the Supplement (as well as in the text), and state this in the text in Section 2.3:

> We assume an uptake coefficient of $\gamma$=0.005 for isoprene nitrates (from both daytime and nighttime chemistry) and $\gamma$=0.01 for all monoterpene nitrates **(Table S4)**.

We now provide more details of the change to absorption cross sections in Section 2.2 and Table S5. We also clarify in the text that the IUPAC recommendation used previously was not for the carbonyl nitrates but for their monofunctional analogues.

> Here we increase the absorption cross sections of the carbonyl INs **following the methodology of Müller et al. (2014, Sect. 2). Briefly, we first use the PROPNN cross section measured by Barnes et al. (1993) to calculate a wavelength-dependent cross section enhancement ratio ($r_{nk}$), defined as the ratio of the measured cross section to the sum of the IUPAC-recommended cross sections for associated monofunctional nitrates and ketones. We then calculate new cross sections for ETHLN, MVKN, and MACRN by multiplying $r_{nk}$ by the sum of cross sections from appropriate monofunctional analogues (Table S5).** The new cross sections are 5-15 times larger than in the original model, which used the IUPAC-recommended **cross section of the monofunctional analogue** tert-butyl nitrate for all carbonyl nitrates (Roberts and Fajer, 1989). **For all species, we calculate photolysis rates assuming unity quantum yields, whereby the weak O−NO2 bond dissociates upon a rearrangement after photon absorption to the carbonyl chromophore (Müller et al., 2014).**

In a couple of places we are re-assured that the kinetic choices provide the best fit to the observational data ("Although simplified, we find this parameterization improves the model fit relative to the SEAC4RS and SOAS . . ...", "In any case, the choice of hydrolysis lifetime does not affect the concentration of gas-phase RONO2 species (because pRONO2 cannot re-partition to the gas phase in the model), and we find this value provides a reasonable match to AMS measurements of total pRONO2 at the surface during SOAS and SEAC4RS"). Could these results in the supplementary material so that the readers can judge these for themselves.

As described above, changes to the model chemistry were added linearly and tested at coarse resolution, so we are unable to add a figure showing "before and after" results for each of these changes. We have edited the text to more clearly reflect that our choices are not necessarily a "best" fit but rather a reasonable fit:

> Although simplified, we find this parameterization **provides a reasonable fit** to the SEAC[4]RS and SOAS observations…"

In both cases, we have added reference to Sect. 3 and 4 where figures showing the fit of the model to observations are shown and results discussed in more detail.

I found sections 3,4,5 rather long for my taste. Would it be possible to put in some sub-section headings to help split those up and allow the reader to find specific bits of information more easily?

We have now split Section 3 into four subsections:
> **3.1 Isoprene and monoterpenes**
> **3.2 First generation $RONO_2$**
> **3.3 Second generation $RONO_2$ and $pRONO_2$**
> **3.4 $RONO_2$-HCHO relationship**

We have now split Section 4 into two subsections:
> **4.1 Speciated versus total $RONO_2$**
> **4.2 $RONO_2$ composition**

Section 5 is less than 2 pages in length, and does not neatly divide into separate sections, so we have left it as is without subsections.

**Anonymous Referee #2**

Hydrolysis is receiving increased attention in chemical transport models as a way to remove organic nitrate (thus NOx) from the atmosphere by converting it to nitric acid. Hydrolysis that has been implemented in (CTMs) chemical transport models thus far (CAMx: Hildebrant Ruiz and Yarwood 2013, CMAQ: Pye et al. 2015) assumes that hydrolysis occurs after semivolatile organics partition to an organic-rich particle. The process of semivolatile organics (monoterpene-derived) undergoing hydrolysis has been demonstrated in laboratory work (e.g. Boyd et al. 2015, Bean and Hildebrandt Ruiz 2016). The timescales for hydrolysis used in CTMs ranges from 3 to 6 hours or longer. Thus, the work of Fisher et al., with a 1 hour hydrolysis represents a significantly faster rate of hydrolysis. Indeed, more than half of the organic nitrates in table 1 have a sub 2-hour lifetime. The irreversible reactive uptake formulation used by Fisher et al. also differs from the equilibrium vapor-pressure based approach. As a result, the Fisher et al. work represents a significantly different approach than has been used thus far for hydrolysis and creation of particulate organic nitrate. The implications for ozone and aerosol may motivate future work to adopt similar formulations and parameter choices should be fully justified.

My main question is whether or not another set of parameters than those examined here would provide equally good model performance. For example, there are no sensitivity simulations to indicate if the 1 hour hydrolysis is significantly better than say a 6 hour hydrolysis. It would be useful to know if another set of conditions (specifically lower BVOC emissions, higher NOx emissions, slower hydrolysis, slower uptake of organic nitrates) also leads to a consistent picture of southeast chemistry. More detailed comments follow.

Throughout our model development and testing, we were unable to find another set of conditions that was consistent with the ensemble of measurements (although that does not mean they do not exist). With respect to the specific set of conditions mentioned:
- Higher $NO_x$ emissions are not consistent with the $SEAC^4RS$ data, as shown in detail in Travis et al. (2016) and discussed below in response to the more detailed comments regarding $NO_x$.
- Lower BVOC emissions would not be consistent with the $SEAC^4RS$ or SOAS observations, as shown in this paper (see also point 6 below regarding BVOC biases). In particular, lower isoprene emissions (already reduced by 15%) would be inconsistent with observed HCHO as described by

Zhu et al. (2016; now available in ACPD), who show a small negative bias of -3% in mixed layer HCHO and an overall negative bias of -10% in total column HCHO.

- Slower (or in some cases, reversible) uptake of organic nitrates is plausible, and we have now made this explicit (see point 4 below). However, slower uptake would significantly degrade model agreement with gas-phase organic nitrates that are currently well represented (e.g., ISOPN, MVKN+MACRN), and this would need to be paired with strengthening (or adding) alternative sinks for these species. The uncertainty of these sinks is already discussed in the context of the ISOPN yield in Section 3.2 ("Including both faster ISOPN photolysis and uptake to the aerosol phase could be a means to accommodate a higher initial ISOPN yield… although both sinks remain unverified. The nature of the sink has implications for $NO_x$ recycling from isoprene nitrates (photolysis recycles $NO_x$ while uptake removes it), and this remains a source of uncertainty in our estimates of the impacts of $RONO_2$ on the $NO_x$ budget.")
- Slower hydrolysis is also plausible. In our current simulation, this would only impact $pRONO_2$ (the additional $HNO_3$ source is minor, as detailed in points 5 and 12 below). We already discuss the implications for $pRONO_2$ in Section 4.2 ("…it is also likely that simulated $pRONO_2$ is underestimated because of our assumption that all $pRONO_2$ species undergo rapid hydrolysis…")

More details on several of these points are provided in response to the detailed comments that follow.

We also note that the short lifetimes of organic nitrates in Table 1 are consistent with an independent analysis of SOAS data by Romer et al. (currently in ACPD), that finds an average lifetime for reactive $RONO_2$ of less than 2 hours. We now cite this work in Section 5.

Travis et al. is referenced as the basis for the gas-phase oxidant chemistry in this work. In that work, they indicate NOx emissions from the EPA NEI are too high by 50%. I assume the simulations in this work (Fisher et al.) reduce NOx emissions consistent with Travis et al.

The reviewer is correct. We now clearly specify this in Section 2 (before 2.1):

> **Our simulation is identical to that used in Travis et al. (2016), Yu et al. (2016), and Zhu et al. (2016).**

Given the suspected importance of reduced NOx emissions, I provide some comments on the Travis et al. paper (available at: http://acmg.seas.harvard.edu/publications/2016/Travis_ACPD_2016.pdf). Reducing the NEI NOx by ~50% may imply a larger hydrolysis sink is needed in the model than is actually necessary. Comments on the Travis et al. paper:

While we are unable to make changes to the Travis et al. (2016) paper – now available at http://www.atmos-chem-phys-discuss.net/acp-2016-110/ – the 50% decrease in NEI NOx applied in that work is well constrained by their observations, and in any case does not inflate the hydrolysis sink used in our work. More details of both points follow.

Travis et al. decrease NEI $NO_x$ on the basis of 2 independent data sets: the SEAC[4]RS airborne data and the NADP nitrate wet deposition fluxes (note that OMI $NO_2$ is not used for this purpose; see below). As stated in their paper (pg. 5, lines 23-24 and pg. 7, lines 2-4 & 13-15):

> "Initial implementation of this [NEI11] inventory in GEOS-Chem resulted in a 60% overestimate of SEAC[4]RS DC-8 observations for $NO_x$ and $HNO_3$, and a 71% overestimate of nitrate ($NO_3^-$) wet deposition fluxes measured by the National Acid Deposition Program (NADP) across the Southeast US… Decreasing [NEI11] emissions corrects the model bias for

NO$_x$ and also largely corrects the bias for inorganic nitrate… the model with decreased NO$_x$ emissions reproduces the spatial variability in the [NADP] observations with minimal bias across the US. In comparison, the model with original emissions had a 60% overestimate of the nitrate wet deposition flux nationally and a 71% overestimate in the Southeast."

These results are further supported by their Fig. 2 and Fig. 3. They also find that the change to NEI emissions removes a 12 ppb bias in boundary layer ozone relative to the SEAC$^4$RS data. Combined, these datasets provide strong evidence for the validity of the NEI decrease applied in the model. The decrease is also consistent with several other studies cited in Travis et al. that find NEI NO$_x$ emissions are too high (Fujita et al., 2012; Brioude et al., 2013; Anderson et al., 2014) .

pRONO$_2$ is positively dependent on surface NO$_X$ emissions, and decreases by ~15% at the surface when NEI emissions are decreased. In other words, with the original NEI emissions, we would require a *faster* hydrolysis sink to maintain the same atmospheric concentration of pRONO$_2$ as in our current model. Note that we determined the most appropriate hydrolysis lifetime on the basis of the pRONO$_2$ concentrations, and impacts on HNO$_3$ are small as described in points 5 and 12 below.

1. Given the missing free tropospheric NO2 in GEOS-Chem and the fact that boundary layer NO2 is less than 30% of the column, how quantitative can the NO2 column evaluation be? What is the error associated with the BEHR and NASA columns? It seems like the GEOS-Chem shape factors may not be a good representation of the NO2 profile given the lack of NO2 aloft. What is the detection limit/precision for surface NO2 from OMI?

Travis et al. use OMI NO$_2$ only to confirm consistency with other evidence, and not as the basis for their NEI emissions reductions. Within the large uncertainties of the satellite retrievals, the OMI data do not cast doubt on the NEI NO$_x$ emissions reductions, and further details of the retrievals are not relevant to our work in this paper.

2. The ratio of ISOPOOH to ISOPN seems degraded with the NOx emission reduction compared to the base indicating too much RO2+HO2 vs RO2+NO. Are there additional later generation products that can be examined to justify proper branching?

We find that the ISOPOOH to ISOPN ratio is not an ideal metric for testing the branching ratio because so many factors besides the branching influence each species, and the uncertainties on the measurements are large (30% for ISOPN, 40% for ISOPOOH).

The NO$_x$ emission decrease improves simulation of isoprene nitrates, with NMB decreasing from +8% to -0.6% for ISOPN and from +23% to -10% for MVKN+MACRN (in both cases, still within the uncertainties).

We note that the ISOPOOH overestimate in the final simulation is paired with an HPALD underestimate, which could indicate that the problems in first-stage isoprene oxidation lie within the low-NO$_x$ pathway rather than the partitioning between low-NO$_x$ and high-NO$_x$ pathways. Both the HPALD underestimate and the ISOPOOH overestimate may be due to kinetic uncertainties, as discussed in Travis et al. (pg. 10, lines 9-16):
"The bias for HPALD is within the uncertainty of the kinetics and measurement. Our HPALD source is based on the ISOPO$_2$ isomerization rate constant from Crounse et al. (2011). A

theoretical calculation by Peeters et al. (2014) suggests a rate constant that is 1.8× higher, which would reduce the model bias for HPALD and ISOPOOH and increase boundary layer OH by 8%. GEOS-Chem overestimates ISOPOOH by 74% below 1.5 km. Recent work by St Clair et al. (2015) found that the reaction rate of ISOPOOH + OH to form IEPOX is approximately 10% faster than the rate given by Paulot et al. (2009b), which would further reduce the model overestimate. It is likely that after these changes the GEOS-Chem overestimate of ISOPOOH would be within measurement uncertainty."

For all three species, GEOS-Chem generally captures the spatial variability (r = 0.8 for ISOPOOH, r=0.7 for HPALD, r=0.6 for ISOPN).

3. How does the magnitude of the soil NOx emission predicted in GEOS-Chem com- pare to other estimates? Does the NEI provide a comparable estimate?

Detailed discussion of the soil $NO_x$ emission is left to the Travis et al. paper. We note that our results are not particularly sensitive to this change. As described in point 12 below, we now include a section in the supplement evaluating the sensitivity of the organic nitrates to the changes in $NO_x$ emissions, including both NEI11 and soil $NO_x$ reductions.

Fisher et al. (this work) comments:

4. Figure S3 indicates too many pRONO2 compared to the AMS data during the day, particularly 13-17 local time. The diurnal profile of the model pRONO2 (relatively flat with peak around 15 local time) also doesn't match the observed AMS profile (peak at night around 3-4 am with minimum around 15 local time). Are the isoprene nitrates too aggressively put in the particle?

This is a plausible explanation, and we have added it to the text at the end of Section 3.3:

**Afternoon overestimates of pRONO2 relative to the AMS observations (Fig. S3) are coincident with the peak in isoprene nitrates (Fig. 6), suggesting overly strong partitioning to the aerosol phase likely due to our assumption of irreversibility (Sect. 2.3).**

5. I recommend performing a total NOy and total inorganic nitrate (HNO3 + NO3-) evaluation. I didn't see a total NOy evaluation in either Travis et al. or Fisher et al. It was a little unclear if Travis et al. included all the updates (ie monoterpenes) of Fisher et al. In any case, Travis shows that HNO3 is high in the base. If the base includes hydrolysis, that may explain why. Furthermore, total RONO2 is underestimated by ~50% according to Figure 10. This implies there are missing NOx sinks (or sources) in the model.

As we now specify in Section 2 (see above), the model runs in this paper and in Travis et al. are identical, and Travis et al. include all the updates from this work.

Travis et al. evaluate total inorganic nitrate (gas $HNO_3$ + aerosol $NO_3^-$) in their Figure 2 (top middle panel, mislabeled as $HNO_3$). The $HNO_3$ increase associated with hydrolysis (~20 ppt) is 20x smaller than the decrease associated with the reduction in $NO_x$ emissions (~400 ppt; see Travis et al. Fig 2). We

now state at the end of Section 2.3 that hydrolysis has minimal impact on simulation of $HNO_3$:

> **Impacts on $HNO_3$ are minor: compared to a simulation without hydrolysis, our simulation with a 1 hr lifetime against hydrolysis increased boundary layer $HNO_3$ by 20 ppt, or 2.4%.**

We have not used total $NO_y$ or inorganic nitrate to constrain our simulation because the measurements are inconsistent with one another. Depending which datasets are used, $\Sigma NO_y$ as measured by the NOAA NOyO3 chemiluminescence instrument differs from the sum of the measured components by as much as 35% (based on median values for surface air over the Southeast US). The experimenters responsible for these measurements have yet to reconcile the differences. Until they do, we do not consider these datasets as appropriate constraints on our simulation of $RONO_2$.

Similarly, measurements of total $RONO_2$ differ by ~50% between $\Sigma ANs$ as measured by TD-LIF and the total of the speciated components as measured by CIT-ToF-CIMS, WAS, and AMS. This is shown in Figure 10a and discussed in the text in Section 4.1. Although we offer some possible explanations in Section 4.1, we cannot arbitrate between the different measurements. Total $RONO_2$ in the model is underestimated relative to the TD-LIF, but not relative to the sum of the speciated measurements, and without further reconciliation of the datasets by the experimenters, we cannot use total $RONO_2$ as a constraint on $NO_x$.

6. Figure 12 shows that as the ratio of BVOC to NOx emissions increases, the fraction of NOx lost to RONO2 (vs HNO3) increases. If BVOC emissions are biased high (Fig 4 shows a NMB of 58% for isoprene and 18% for monoterpenes compared to DC8 flights) and NOx emissions are biased low (2011 NEI is reduced ~50%) then the model will predict too much NOx lost to RONO2 vs HNO3. Hydrolysis converts RONO2 to HNO3. Another set of BVOC and NOx emission levels and their evaluation would be useful.

It would be nice to see analysis examining the possibility of lower BVOC emissions, higher NOx emissions, slower hydrolysis, and slower uptake of organic nitrates leading to a consistent picture of southeast chemistry.

Upon reading this comment, we realised that the normalised mean bias (NMB) values given in Fig. 4 were misleading. The NMB was calculated from all data below 1 km, but as seen in Fig. 5 there are discrepancies in the simulated profile shape above 500 m that inflate the bias at the surface. The NMB for data below 500 m is more representative for the purpose of evaluating emissions, and we now provide this instead. Because the figure shows data below 1 km but the NMB is calculated for data below 500 m, we have removed the NMB from the figure itself and instead state it in the caption:

> **The normalized mean bias of the simulation relative to the PTR-MS measurements in the lowest 500 m is +34% for isoprene and +3% for monoterpenes.**

We have similarly updated Fig. 7.

Although there is still a positive bias for isoprene, we have already decreased isoprene emissions by 15% relative to the standard MEGAN model and it is not clear that even lower emissions would be consistent with the ensemble of measurements. We show in Fig. 6 that isoprene is biased low relative to the SOAS data, and Zhu et al. (2016) show that mixed layer HCHO currently has a very small negative bias (-3% ± 2%) relative to the aircraft observations. The justification for the reduction in NEI 2011 $NO_x$ emissions is discussed above. We do not feel that another set of emissions is justified.

In any case, our finding that $NO_x$ loss to $RONO_2$ increases with lower $NO_x$ emissions is consistent with the findings of Browne and Cohen (2012). They analysed an environment with different BVOC and $NO_x$ emissions (boreal Canada) and did not include aerosol uptake or subsequent hydrolysis in their model – suggesting this result is not particularly sensitive to these uncertainties.

7. Page 7, line 32: The yield of organic nitrates from API+NO3 is set to 10%. This seems to be on the low end considering the values tabulated in Fry et al. 2014 which range from 11-29% for a-pinene and 40-74% for b-pinene. Is there a specific RO2 fate assumed for these yields?

We applied the yield from the RACM-2 mechanism on which our mechanism was built (Goliff et al., 2013; Browne et al., 2014). As the resultant MONIT was already higher than observed, we did not update this value. We have now clarified in Section 2.3 that this should be considered a lower bound:

> The branching ratio between these two fates is 50% nitrate-retaining for $LIM + NO_3$ (Fry et al., 2014) and 10% nitrate-retaining for $API + NO_3$ (Browne et al., 2014). **The 10% nitrate yield from $API + NO_3$ is on the low end of the observed range (Fry et al., 2014), so simulated pinene-derived MONIT should be considered a lower bound.**

Relevant reactions including rates and products are now shown in the Supplement.

8. Page 8, line 33: Is there a figure that can be referenced to show good RONO2 performance?

As described in the response to Reviewer 1, changes to the model chemistry were added linearly and tested at coarse resolution, so we are unable to add a figure showing "before and after" results for this change. We have edited the text to more clearly state this:

> Although simplified, we find this parameterization **provides a reasonable fit** to the SEAC[4]RS and SOAS observations…"

We have also added a reference to Sect. 3 and 4 where figures showing the fit of the model to observations are shown and results discussed in more detail.

9. Page 9, near line 7, how was the specific lifetime of 1 hour against hydrolysis chosen? Were other values used in preliminary simulations? For the isoprene system, what is the fraction of tertiary vs. non-tertiary nitrates?

We now specify in Section 2.3:

> **We assume here a bulk lifetime against hydrolysis of 1 hr, which we found in preliminary simulations to provide a better simulation of $pRONO_2$ than longer lifetimes.**

We were unable to find estimates of the fraction of tertiary vs. non-tertiary nitrates under atmospheric conditions. For first generation isoprene nitrates, the tertiary $\beta(1,2)$ isomer accounts for ~23% of the yield at the time of formation (Paulot et al., 2009). For second generation isoprene nitrates, the SEAC[4]RS observations (Fig. 5) indicate a dominant contribution from MVKN (non-tertiary) + MACRN (tertiary), but cannot distinguish between the two. At this stage, for isoprene nitrates we expect the uncertainty in the uptake parameterization (see point 4 above) has a larger impact on the simulation than our choice of hydrolysis lifetime (which would change with a different uptake parameterization).

10. Page 9, line 16, in reality, the hydrolysis lifetime should affect gas-phase RONO2. In the reactive uptake formulation of Marais et al. the uptake coefficient depends on the rate of particle-phase reaction (here hydrolysis). Faster reaction increases the uptake coefficient. It should be pointed out that this dependence of gamma on hydrolysis is not captured by the model.

We now state this in Section 2.3:
> In other words, our implementation of aerosol partitioning involves a two-step process of (1) uptake of gas- phase $RONO_2$ to form a simplified non-volatile $pRONO_2$ species, with rate determined by $\gamma$, followed by (2) hydrolysis of the simplified $pRONO_2$ species to form $HNO_3$, with rate determined by the lifetime against hydrolysis. **These steps are de-coupled, and we do not include any dependence of $\gamma$ on the hydrolysis rate (unlike the more detailed formulation of Marais et al. (2016)).**

11. Page 13: Are the TD-LIF measurements of total ANs consistent with the SEARCH network measurements during SOAS?

We now include the following at the end of the first paragraph in Section 4.1:
> **An independent thermal dissociation instrument operated by the SouthEastern Aerosol Research and Characterization (SEARCH) Network also measured $\Sigma$ANs at the SOAS site and showed values that were 80 ppt higher than measured by the TD-LIF (but generally well correlated, with slope close to 1 and $r \sim 0.8$).**

12. A few things indicate the NOx emissions may have been reduced too much:
o the total ANs are low by ~50% (Figure 10)
o The ratio of ISOPOOH to ISOPN seems degraded (too high) with the NOx emission reduction compared to the base (Travis et al. Figure 2)
o The NO2 columns are low compared to observations (Travis et al.) and the ability of the satellite products to determine the magnitude of a given error in surface NO2 is unclear
o A source of HNO3 (hydrolysis) must be added to the model

The points relating to ISOPOOH/ISOPN and $NO_2$ columns are addressed above.

With respect to the total ANs underestimate, in addition to existing text discussing the discrepancy (Section 4.1, see above) we now include a new section in the Supplement:
> **S1. $RONO_2$ sensitivity to $NO_X$ emissions reductions**
> **We use the same simulation as Travis et al. (2016), who reduced $NO_x$ emissions in the NEI11v1 inventory by 60% for all anthropogenic sources except power plants (equivalent to a 53% decrease in total annual NEI11v1 emissions) and also reduced soil $NO_x$ emissions in the Midwest US by 50% (Vinken et al., 2014). Figure S5 compares model results during SEAC$^4$RS before and after applying these $NO_x$ emissions decrease. As seen in the figure, the change to $NO_x$ emissions cannot explain the model underestimate in $\Sigma$ANs relative to the SEAC$^4$RS TD-LIF measurement (-46% with original $NO_x$, -57% with reduced $NO_x$). The figure also shows that the change to $NO_x$ emissions does not have an appreciable effect on simulation of individual $RONO_2$ species, which fall within the experimental uncertainties of the CIT-ToF-CIMS instrument in both versions of the model.**

The new Figure S5 can be found at the end of this document.

With respect to $HNO_3$, the hydrolysis was added to provide a sink for $pRONO_2$ rather than to provide a source of $HNO_3$. In fact, hydrolysis has no appreciable impact on $HNO_3$, and we now state this at the end of Section 2.3:

> **Impacts on $HNO_3$ are minor: compared to a simulation without hydrolysis, our simulation with a 1 hr lifetime against hydrolysis increased boundary layer $HNO_3$ by 20 ppt, or 2.4%.**

13. In addition to posting the mechanism online, can full details be provided in the supporting material for future reference?

As detailed in the response to Reviewer 1, we now provide the full monoterpene nitrate mechanism in the Supplement. Updates to the isoprene oxidation mechanisms are provided in the Supplement to Travis et al.

14. While I am aware of studies of isoprene nitrates undergoing hydrolysis in bulk systems (e.g. Jacobs et al. 2014 ACP), what studies explore the interaction of gas-phase isoprene nitrates and aerosol-phase hydrolysis products for the isoprene system?

As far as we are aware the only study that begins to address these questions for the isoprene nitrates is Lee et al. (2016) – not yet published when we were developing our model. This is why many of our parameters (e.g., uptake coefficients, hydrolysis lifetimes) were selected as best guesses to fit the species for which we did have observational constraints.

References

Bean, J. K. and Hildebrandt Ruiz, L.: Gas–particle partitioning and hydrolysis of organic nitrates formed from the oxidation of α-pinene in environmental chamber experiments, Atmos. Chem. Phys., 16, 2175-2184, doi:10.5194/acp-16-2175-2016, 2016.

Hildebrandt Ruiz, L.; Yarwood, G. Interactions between organic aerosol and NOy: influence on oxidant production, AQRP Project 12–012; Report, 2013, aqrp.ceer.utexas.edu/projectinfoFY12_13/12-012/12-012%20Final%20Report.pdf.

Jacobs, M. I., Burke, W. J., and Elrod, M. J.: Kinetics of the reactions of isoprene- derived hydroxynitrates: gas phase epoxide formation and solution phase hydrolysis, Atmos. Chem. Phys., 14, 8933-8946, doi:10.5194/acp-14-8933-2014, 2014.

[Figure]

**Figure S5. Observed (black) and modeled (red) median 0-4 km profiles of RONO₂ over the Southeast US during SEAC⁴RS. The dotted red line shows model results before scaling non-power plant NOₓ emissions from the NEI11v1 inventory and soil NOₓ in the Midwest US.**

**References (*only those not in original manuscript or updated since submission*)**

Anderson, D. C., Loughner, C. P., Diskin, G., Weinheimer, A., Canty, T., P., Salawitch, R. J., Worden, H. M., Fried, A., Mikoviny, T., Wisthaler, A., and Dickerson, R., R.: Measured and modeled CO and NOy in DISCOVER-AQ: An evaluation of emissions and chemistry over the eastern US, Atmospheric Environment, 96, 78-87, doi:10.1016/j.atmosenv.2014.07.004, 2014.

Barnes, I., Becker, K. H., and Zhu, T.: Near UV absorption spectra and photolysis products of difunctional organic nitrates: Possible importance as NO x reservoirs, Journal of Atmospheric Chemistry, 17, 353–373, doi:10.1007/BF00696854, 1993.

Brioude, J., Angevine, W. M., Ahmadov, R., Kim, S. W., Evan, S., McKeen, S. A., Hsie, E. Y., Frost, G. J., Neuman, J. A., Pollack, I. B., Peischl, J., Ryerson, T. B., Holloway, J., Brown, S. S., Nowak, J. B., Roberts, J. M., Wofsy, S. C., Santoni, G. W., Oda, T., and Trainer, M.: Top-down estimate of surface flux in the Los Angeles Basin using a mesoscale inverse modeling technique: assessing anthropogenic emissions of CO, NOx and CO2 and their impacts, Atmospheric Chemistry and Physics, 13, 3661-3677, doi:10.5194/acp-13-3661-2013, 2013.

Fujita, E. M., Campbell, D. E., Zielinska, B., Chow, J. C., Lindhjem, C. E., DenBleyker, A., Bishop, G. A., Schuchmann, B. G., Stedman, D. H., and Lawson, D. R.: Comparison of the MOVES2010a, MOBILE6.2, and EMFAC2007 mobile source emission models with on-road traffic tunnel and remote sensing measurements, Journal of the Air & Waste Management Association, 62, 1134-1149, doi:10.1080/10962247.2012.699016, 2012.

Lee, B. H., Mohr, C., Lopez-Hilfiker, F. D., Lutz, A., Hallquist, M., Lee, L., Romer, P., Cohen, R. C., Iyer, S., Kurten, T., Hu, W. W., Day, D. A., Campuzano-Jost, P., Jimenez, J. L., Xu, L., Ng, N. L., Guo, H., Weber, R. J., Wild, R. J., Brown, S. S., Koss, A., de Gouw, J., Olson, K., Goldstein, A. H., Seco, R., Kim, S., McAvey, K. M., Shepson, P. B., Starn, T., Baumann, K., Edgerton, E., Liu, J., Shilling, J. E., Miller, D. O., Brune, W. H., Schobesberger, S., D'Ambro, E. L., and Thornton, J. A.: Highly functionalized organic nitrates in the Southeast U.S.: contribution to secondary organic aerosol and reactive nitrogen budgets, Proceedings of the National Academy of Sciences, 113, 1516–1521, doi:10.1073/pnas.1508108113, 2016.

Marais, E. A., Jacob, D. J., Jimenez, J. L., Campuzano-Jost, P., Day, D. A., Hu, W., Krechmer, J., Zhu, L., Kim, P. S., Miller, C. C., Fisher, J. A., Travis, K., Yu, K., Hanisco, T. F., Wolfe, G. M., Arkinson, H. L., Pye, H. O. T., Froyd, K. D., Liao, J., and McNeill, V. F.: Aqueous- phase mechanism for secondary organic aerosol formation from isoprene: application to the Southeast United States and co-benefit of SO2 emission controls, Atmospheric Chemistry and Physics, 16, 1603–1618, doi:10.5194/acp-16-1603-2016, 2016.

Romer, P. S., Duffey, K. C., Wooldridge, P. J., Allen, H. M., Ayres, B. R., Brown, S. S., Brune, W. H., Crounse, J. D., de Gouw, J., Draper, D. C., Feiner, P. A., Fry, J. L., Goldstein, A. H., Koss, A., Misztal, P. K., Nguyen, T. B., Olson, K., Teng, A. P., Wennberg, P. O., Wild, R. J., Zhang, L., and Cohen, R. C.: The Lifetime of Nitrogen Oxides in an Isoprene Dominated Forest, Atmospheric Chemistry and Physics Discussions, 2016, 1–25, doi:10.5194/acp-2016-28, 2016.

Travis, K. R., Jacob, D. J., Fisher, J. A., Kim, P. S., Marais, E. A., Zhu, L., Yu, K., Miller, C. C., Yantosca, R. M., Sulprizio, M. P., Thompson, A. M., Wennberg, P. O., Crounse, J. D., St. Clair, J. M., Cohen, R. C., Laughner, J. L., Dibb, J. E., Hall, S. R., Ullmann, K., Wolfe, G. M., Neuman, J. A., and Zhou, X.: NOx emissions, isoprene oxidation pathways, vertical mixing, and implications for surface ozone in the Southeast United States, in review for Atmospheric Chemistry and Physics, 2016.

Vinken, G. C. M., Boersma, K. F., Maasakkers, J. D., Adon, M., and Martin, R. V.: Worldwide biogenic soil NOx emissions inferred from OMI NO2 observations, Atmospheric Chemistry and Physics, 14, 10363-10381, doi:10.5194/acp-14- 10363-2014, 2014.

Zhu, L., Jacob, D. J., Kim, P. S., Fisher, J. A., Yu, K., Travis, K. R., Mickley, L. J., Yantosca, R. M., Sulprizio, M. P., De Smedt, I., Gonzalez Abad, G., Chance, K., Li, C., Ferrare, R., Fried, A., Hair, J. W., Hanisco, T. F., , Richter, D., Scarino, A. J., Walega, J., Weibring, P., and Wolfe, G. M.: Observing atmospheric formaldehyde (HCHO) from space: validation and intercomparison of six retrievals from four satellites (OMI, GOME2A, GOME2B, OMPS) with SEAC4RS aircraft observations over the Southeast US, in review for Atmospheric Chemistry and Physics, 2016.

---

## Author Comment (AC3) · 13 Apr 2016

**S1. RONO$_2$ sensitivity to NO$_x$ emissions reductions**

We use the same simulation as Travis et al. (2016), who reduced NO$_x$ emissions in the NEI11v1 inventory by 60% for all anthropogenic sources except power plants (equivalent to a 53% decrease in total annual NEI11v1 emissions) and also reduced soil NO$_x$ emissions in the Midwest US by 50% (Vinken et al., 2014). Figure S5 compares model results during SEAC$^4$RS before and after applying these NO$_x$ emissions decreases. As seen in the figure, the change to NO$_x$ emissions cannot explain the model underestimate in ΣANs relative to the SEAC$^4$RS TD-LIF measurement (-46% with original NO$_x$, -57% with reduced NO$_x$). The figure also shows that the change to NO$_x$ emissions does not have an appreciable effect on simulation of individual RONO$_2$ species, which fall within the experimental uncertainties of the CIT-ToF-CIMS instrument in both versions of the model.

[Figure]

**Figure S1.** Comparison of median vertical profiles from PTR-MS (solid black) and Whole Air Sampler (WAS) measurements of isoprene and monoterpenes (=α-pinene+β-pinene for the WAS) over the Southeast US during SEAC$^4$RS (left). The observations are also compared to the GEOS-Chem simulation sampled in the same manner as each measurement in the center (PTR-MS) and right (WAS) panels.

[Figure]

**Figure S2.** Same as Fig. 4, but for NO$_X$.

[Figure]

**Figure S3.** Same as Fig. 6, but for individual second generation isoprene nitrates (MVKN+MACRN, PROPNN, ETHLN) and summed first generation monoterpene nitrates (MONIT) from the CIT-ToF-MS (1 June – 4 July), particulate organic nitrates (pRONO$_2$) from the AMS (9 June – 15 July) and the TD-LIF (29 June – 15 July), and total organic nitrates (ΣANs) from the TD-LIF (1 June – 15 July).

[Figure]

**Figure S4.** Same as Fig. 8, but for ISOPN measured by CIT-ToF-MS and for pRONO₂ measured by TD-LIF.

**a. Total RONO₂ (ΣAN)**

- —— SEAC⁴RS ΣAN
- —— GEOS-Chem
- ······ GEOS-Chem, original NOₓ

**b. ISOPN**

**c. MVKN + MACRN**

**d. PROPNN**

**e. ETHLN**

Mixing Ratio, ppt

**Figure S5.** Observed (black) and modeled (red) median 0-4 km profiles of RONO₂ over the Southeast US during SEAC⁴RS. The dotted red line shows model results before scaling non-power plant NOₓ emissions from the NEI11v1 inventory and soil NOₓ in the Midwest US.

**Table S1.** New species added to GEOS-Chem for monoterpene nitrate chemistry.

| Abbreviation | Name |
| --- | --- |
| API | alpha-pinene and other cyclic terpenes with one double bond |
| APIO2 | $RO_2$ from API |
| LIM | limonene and other cyclic terpenes with two double bonds |
| LIMO2 | $RO_2$ from LIM |
| PIP | peroxides from API & LIM |
| OLND | monoterpene-derived NO3-alkene adduct that primarily decomposes |
| OLNN | monoterpene-derived NO3-alkene adduct that primarily retains the NO3 functional group |
| MONITS | saturated first generation monoterpene organic nitrate |
| MONITU | unsaturated first generation monoterpene organic nitrate |
| HONIT | second generation monoterpene organic nitrate |

**Table S2.** New kinetic reactions added to GEOS-Chem for monoterpene nitrate chemistry.

| Reactants | Products | Rate Constant |
| --- | --- | --- |
| API + OH | APIO2 | 1.21E-11*exp(440/T) |
| APIO2 + NO | 0.82HO2 + 0.82NO2 + 0.23HCHO+ 0.43 RCHO + 0.11 ACET + 0.44MEK + 0.07 HCOOH + 0.12MONITS + 0.06MONITU | 4.00E-12 |
| APIO2 + HO2 | PIP | 1.50E-11 |
| APIO2 + MO2 | HO2 + 0.75HCHO + 0.25 MOH + 0.25 ROH + 0.75RCHO + 0.75MEK | 3.56E-14*exp(708/T) |
| APIO2 + MCO3 | 0.5 HO2 + 0.5 MO2 + RCHO + MEK + RCOOH | 7.40E-13*exp(765/T) |
| APIO2 + NO3 | HO2 + NO2 + RCHO + MEK | 1.20E-12 |
| API + O3 | 0.85OH + 0.1HO2 + 0.62 KO2 + 0.14 CO + 0.02 H2O2 + 0.65RCHO + 0.53MEK | 5.0E-16*exp(-530/T) |
| APIO2 + NO3 | 0.1OLNN + 0.9 OLND | 8.33E-13*exp(490/T) |
| LIM + OH | LIMO2 | 4.20E-11*exp(401/T) |
| LIMO2 + NO | 0.686HO2 + 0.78NO2 + 0.22MONITU + 0.289 PRPE + 0.231HCHO + 0.491RCHO + 0.058HAC + 0.289MEK | 4.00E-12 |
| LIMO2 + HO2 | PIP | 1.50E-11 |
| LIMO2 + MO2 | HO2 + 0.192 PRPE + 1.04 HCHO + 0.308 MACR + 0.25 MOH + 0.25 ROH | 3.56E-14*exp(708/T) |
| LIMO2 + MCO3 | 0.5 HO2 + 0.5 MO2 + 0.192PRPE + 0.385 HCHO + 0.308 MACR + 0.5 RCOOH | 7.40E-13*exp(765/T) |
| LIMO2 + NO3 | HO2 + NO2 + 0.385PRPE + 0.385HCHO + 0.615MACR | 1.20E-12 |
| LIM + O3 | 0.85OH + 0.10HO2 + 0.16 ETO2 + 0.42 KO2 + 0.02H2O2 + 0.14CO + 0.46PRPE + 0.04HCHO + 0.79MACR + | 2.95E-15*exp(-783/T) |

| | 0.01HCOOH + 0.07 RCOOH | |
|---|---|---|
| LIM + NO3 | 0.5OLNN + 0.5OLND | 1.22E-11 |
| PIP + OH | 0.49OH + 0.44R4O2 + 0.08RCHO + 0.41MEK | 3.4E-12*exp(190/T) |
| OLNN + NO | HO2 + NO2 + MONITS | 4.00E-12 |
| OLND + NO | 2.0 NO2 + 0.287 HCHO + 1.24 RCHO + 0.464 MEK | 4.00E-12 |
| OLNN + HO2 | 0.7MONITS + 0.3MONITU | 1.66E-13*exp(1300/T) |
| OLND + HO2 | 0.7MONITS + 0.3MONITU | 1.66E-13*exp(1300/T) |
| OLNN + MO2 | 2.0 HO2 + HCHO + 0.7MONITS + 0.3MONITU | 1.60E-13*exp(708/T) |
| OLND + MO2 | 0.5 HO2 + 0.5 NO2 + 0.965 HCHO + 0.93 RCHO + 0.348 MEK + 0.25 MOH + 0.25 ROH + 0.35 MONITS + 0.15 MONITU | 9.68E-14*exp(708/T) |
| OLNN + MCO3 | HO2 + MO2 + 0.7 MONITS + 0.3 MONITU | 8.85E-13*exp(765/T) |
| OLND + MCO3 | 0.5MO2 + NO2 + 0.287 HCHO + 1.24 RCHO + 0.464 MEK + 0.5 RCOOH | 5.37E-13*exp(765/T) |
| OLNN + NO3 | HO2 + NO2 + 0.7 MONITS + 0.3 MONITU | 1.20E-12 |
| OLND + NO3 | 2.0NO2 + 0.287 HCHO + 1.24 RCHO + 0.464 MEK | 1.20E-12 |
| OLNN + OLNN | HO2 + 1.4 MONITS + 0.6 MONITU | 7.0E-14*exp(1000/T) |
| OLNN + OLND | 0.5 HO2 + 0.5 NO2 + 0.202HCHO + 0.64 RCHO + 0.149 MEK +1.05 MONITS + 0.45 MONITU | 4.25E-14*exp(1000/T) |
| OLND + OLND | NO2 + 0.504 HCHO + 1.21 RCHO + 0.285MEK + 0.7 MONITS + 0.3 MONITU | 2.96E-14*exp(1000/T) |
| MONITS + OH | HONIT | 4.80E-12 |
| MONITU + OH | HONIT | 7.29E-11 |
| MONITU + O3 | HONIT | 1.67E-16 |
| MONITU + NO3 | HONIT | 3.15E-13*exp(-448/T) |
| MONITS + NO3 | HONIT | 3.15E-13*exp(-448/T) |
| HONIT + OH | NO3 + HKET | same as HNO3 + OH |

**Table S3.** New photolysis reactions added to GEOS-Chem for monoterpene nitrate chemistry

| Species | Photolysis Products | j-value used |
|---|---|---|
| PIP | OH + HO2 + RCHO | j(H2O2) |
| MONITS | MEK + NO2 | j(ONIT1) |
| MONITU | RCHO + NO2 | j(ONIT1) |
| HONIT | HKET + NO2 | j(ONIT1) |

**Table S4.** Reactive uptake coefficients (γ) used in GEOS-Chem aerosol uptake parameterization for organic nitrates. [a]

| Species | γ |
|---|---|
| ISOPNB | 0.005 |
| ISOPND | 0.005 |
| ISN1 | 0.005 |
| INPN | n/a |
| MVKN | 0.005 |
| MACRN | 0.005 |
| PROPNN | n/a |
| ETHLN | n/a |
| R4N2 | 0.005 |
| DHDN | 0.005 |
| MONITS | 0.01 |
| MONITU | 0.01 |
| HONIT | 0.01 |

[a] For full species names, see Table 1. Species that do not partition to the aerosol are indicated with "n/a".

**Table S5.** Monofunctional analogues used in calculation of updated absorption cross sections for carbonyl nitrates. [a]

| Species | Ketone analogue | Nitrate analogue |
|---|---|---|
| ETHLN | ethanal [b] | ethyl nitrate [b] |
| MACRN | i-butyraldehyde [c] | tert-butyl nitrate [d] |
| MVKN | 2-butanone [b] | 2-butyl nitrate [b] |

[a] Wavelength-dependent cross sections for carbonyl nitrates are calculated following Müller et al. (2014) by first calculating the PROPNN cross section enhancement ratio $r_{nk}(\lambda) = S_{nk}(\lambda)/(S_n(\lambda)+S_k(\lambda))$, where $S_{nk}(\lambda)$ is the PROPNN cross section measured by Barnes et al. (1993) and $S_n(\lambda)$ and $S_k(\lambda)$ are the cross sections of the associated monofunctional nitrate and ketone, respectively. Cross sections for other carbonyl nitrates are calculated by applying the PROPNN enhancement ratio to the cross sections of the appropriate monofunctional ketones and nitrates as given in the table: $S(\lambda) = r_{nk}(\lambda)[S_n(\lambda)+S_k(\lambda)]$.
[b] Atkinson et al. (2006)
[c] Martinez et al. (1992)
[d] Roberts and Fajer (1989)

**References**

Atkinson, R., Baulch, D. L., Cox, R. A., Crowley, J. N., Hampson, R. F., Hynes, R. G., Jenkin, M. E., Rossi, M. J., Troe, J., and Subcommittee, I.: Evaluated kinetic and photochemical data for atmospheric chemistry: Volume II ndash; gas phase reactions of organic species, Atmo- spheric Chemistry and Physics, 6, 3625–4055, doi:10.5194/acp-6-3625-2006, 2006.

Barnes, I., Becker, K. H., and Zhu, T.: Near UV absorption spectra and photolysis products of difunctional organic nitrates: Possible importance as NO x reservoirs, Journal of Atmospheric Chemistry, 17, 353–373, doi:10.1007/BF00696854, 1993.

Martinez, R. D., Buitrago, A. A., Howell, N. W., Hearn, C. H., and Joens, J. A.: The near U.V. absorption spectra of several aliphatic aldehydes and ketones at 300 K, Atmospheric Environment. Part A. General Topics, 26, 785 – 792, doi: 10.1016/0960-1686(92)90238-G, 1992.

Müller, J.-F., Peeters, J., and Stavrakou, T.: Fast photolysis of carbonyl nitrates from isoprene, Atmospheric Chemistry and Physics, 14, 2497–2508, doi:10.5194/acp-14-2497-2014, 2014.

Roberts, J. M. and Fajer, R. W.: UV absorption cross sections of organic nitrates of potential atmospheric importance and estimation of atmospheric lifetimes, Environmental Science & Technology, 23, 945–951, doi:10.1021/es00066a003, 1989.

Travis, K. R., Jacob, D. J., Fisher, J. A., Kim, P. S., Marais, E. A., Zhu, L., Yu, K., Miller, C. C., Yantosca, R. M., Sulprizio, M. P., Thompson, A. M., Wennberg, P. O., Crounse, J. D., St. Clair, J. M., Cohen, R. C., Laughner, J. L., Dibb, J. E., Hall, S. R., Ullmann, K., Wolfe, G. M., Neuman, J. A., and Zhou, X.: NOx emissions, isoprene oxidation pathways, vertical mixing, and implications for surface ozone in the Southeast United States, in review for Atmospheric Chemistry and Physics, 2016.